



# Evaluation of modelled versus observed NMVOC compounds at EMEP sites in Europe

Yao Ge[1,2], Sverre Solberg[3], Mathew Heal[4], Stefan Reimann[5], Willem van Caspel[1], Bryan Hellack[6], Thérèse Salameh[7], and David Simpson[1,8]

[1]Division for Climate Modelling and Air Pollution, Norwegian Meteorological Institute, 0313 Oslo, Norway
[2]UK Centre for Ecology & Hydrology, Bush Estate, Penicuik, EH26 0QB, UK
[3]NILU, 2027 Kjeller, Norway
[4]School of Chemistry, University of Edinburgh, Joseph Black Building, Edinburgh, EH9 3FJ, UK
[5]Empa - Laboratory for Air Pollution/Environmental Technology, Swiss Federal Laboratories for Materials Science and Technology, CH-8600 Dübendorf, Switzerland.
[6]German Environment Agency, Dessau, Germany
[7]IMT Nord Europe, Institut Mines-Télécom, Univ. Lille, Centre for Energy and Environment, 59000, Lille, France
[8]Department of Space, Earth and Environment, Chalmers University of Technology, 412 96 Gothenburg, Sweden

**Correspondence:** Yao Ge (yaog@met.no) and David Simpson (davids@met.no)



# 1  Abstract

Atmospheric volatile organic compounds (VOC) constitute a wide range of species, acting as precursors to ozone and aerosol formation. Atmospheric chemistry and transport models (CTMs) are crucial to understanding the emissions, distribution, and impacts of VOCs. Given the uncertainties in VOC emissions, lack of evaluation studies, and recent changes in emissions, this work adapts the European Monitoring and Evaluation Programme Meteorological Synthesizing Centre – West (EMEP MSC-W) CTM to evaluate emission inventories in Europe. Here we undertake the first intensive model-measurement comparison of VOCs in two decades. The modelled surface concentrations are evaluated both spatially and temporally, using measurements from the regular EMEP monitoring network in 2018 and 2019, and a 2022 campaign. To achieve this, we utilised the UK National Atmospheric Emission Inventory to derive explicit emission profiles for individual species and employed a 'tracer' method to produce pure concentrations that are directly comparable to observations. Model simulations for 2018 compare the use of two European inventories, CAMS and CEIP, and of two chemical mechanisms, CRIv2R5Em and EmChem19rc; those for 2019 and 2022 use CAMS and CRIv2R5Em only.

The degree to which the modelled and measured VOCs agree varies depending on the specific species. The model successfully captures the overall spatial and temporal variations of major alkanes (e.g., ethane, n-butane) and unsaturated species (e.g., ethene, benzene), but less though for propane, i-butane, and ethyne. This discrepancy underscores potential issues in the boundary conditions for these latter species and in their primary emissions from in particular the solvent and road transport sectors. Specifically, potential missing propane emissions and issues with its boundary conditions are highlighted by large model underestimations and smaller propane to ethane ratios compared to the measurement. Meanwhile, both the model and measurement show strong linear correlations among butane isomers and among pentane isomers, indicating common sources for these pairs of isomers. However, modelled ratios of i- to n-butane and i- to n-pentane are approximately one-third of the measured ratios, which is largely driven by significant emissions of n-butane and n-pentane from the solvent sector. This suggests issues with the speciation profile of the solvent sector, or underrepresented contributions from transport and fuel evaporation sectors in current inventories, or both. Furthermore, the modelled ethene-to-ethyne and benzene-to-ethyne ratios differ significantly from measured ratios. The different model performance strongly points to shortcomings in the spatial and temporal patterns and magnitudes of ethyne emissions, especially during winter. For OVOCs, modelled and measured methanal and methylglyoxal display a good agreement, which demonstrates that the model captures the overall photo-oxidation processes reasonably well. However, the insufficiency of suitable measurements limits the evaluation of other OVOCs. Finally, the model exhibits very similar performance across simulations using different inventories, which suggests that the emission profiles are likely to exert a more significant impact on the agreement between modelled and measured data than the total emissions reported for each sector. Therefore, the future focus may need to shift towards refining these speciation profiles through for example new emission measurement campaigns to improve the model accuracy.





## 2  Introduction

Non-methane volatile organic compounds (NMVOCs) constitute a diverse category of organic chemicals. While only a limited number of VOCs emitted to air are known to be directly detrimental to health, they predominantly serve as precursors to the formation of ozone and particulate matter (PM) (Seinfeld and Pandis, 1998; Ait-Helal et al., 2014; Li et al., 2020; Pye et al., 2022). Upon their release into the atmosphere, VOCs undergo a series of photochemical reactions that lead to the generation of ground-level ozone that has well-known adverse effects on air quality, human health, crops and natural vegetation (Filleul et al., 2006; Hoor et al., 2009; Mills et al., 2018; Li et al., 2019; Emberson, 2020). Concurrently, VOCs also affect the mass, number, and chemical composition of PM through their contributions to primary and secondary organic aerosols (Kanakidou et al., 2005; Kroll and Seinfeld, 2008; Hallquist et al., 2009). Consequently, the reduction of VOC levels remains a critical factor in mitigating both surface ozone and PM pollution.

The spatial and temporal concentrations of VOCs are influenced by a range of atmospheric processes. These include primary emissions from a number of sources, chemical transformations, regional transport, and variations in meteorological conditions. Further, difficulties in emissions estimation and model parameterisation of these processes, combined with technical challenges in accurately measuring ambient speciated VOC levels, often leads to varying agreements between model and measurement (Solberg et al., 2001; Pfister et al., 2008; Veefkind et al., 2012; Huang et al., 2017; Dalsøren et al., 2018; Bray et al., 2019; von Schneidemesser et al., 2023).

Comparison of modelled VOC results with observations presents a number of challenges beyond those of other compounds such as NOx, SOx or NH$_3$ whose emissions are compiled annually. In contrast, the regular assessment of speciated emission data for VOCs is much rarer. In many cases, emissions are input to an CTM as total non-methane VOC, but these emissions then need to be converted to inputs of specific species (C$_2$H$_6$, C$_3$H$_8$ etc) because each VOC has different ozone and aerosol formation potential, and few data are available to support this speciation. Thus, results from models can diverge significantly based on different VOC emission profiles used. Also, the lifetime of many VOCs is so short that a sound comparison of measured and modelled levels is difficult. Moreover, a particular monitoring site's representativeness of its surrounding air, and the quality of its measurement data, can also vary dramatically. In response to these challenges, continuous efforts have been invested in implementing long-term VOC measurements across Europe (e.g., the pan-European research infrastructure ACTRIS) and enhancing chemistry mechanisms in the European Monitoring and Evaluation Programme Meteorological Synthesizing Centre – West (EMEP MSC-W) CTM in recent years.

A further crucial issue is that real-world VOCs comprise many 1000s of species, but chemical transport model schemes can only cope with a much smaller number of compounds, typically in the 100s. In the default chemistry mechanisms of the EMEP model, EmChem19a (Simpson et al., 2020; Bergström et al., 2022), and the update, EmChem19rc, most emitted VOCs are lumped into different groups (e.g. most alkanes are treated as n-butane), with only a few VOC species having explicit emissions and chemistry. This approach offers the dual benefit of maintaining an accurate description of ozone generation compared to more complex schemes (Andersson-Sköld and Simpson, 1999; Bergström et al., 2022) and promoting computational efficiency. However, it presents challenges when attempting to produce specific VOC concentrations for comparison with observation data.



In recent decades, considerable declines have been observed in those VOCs that are primarily derived from transport, com-
bustion and fossil fuel extraction and distribution (sectors that were dominant emission sources in the 1990s) due to changes
in emission regulation and fuel quality, as well as increases in the usage of renewable energy. Meanwhile, emissions from
solvents, and the use of chemicals in industry and domestic products, as well as other sources like agricultural activities, are
70 gaining in significance (von Schneidemesser et al., 2016; Mo et al., 2021). These changes in major emitting sectors have con-
sequently led to changes in VOC emission profiles. For instance, there has been a notable reduction in emissions of short-chain
non-methane hydrocarbons (NMHCs) associated with fossil fuels and combustion, and an increase in the relative contributions
of oxygenated VOCs (OVOCs) which primarily emanate from solvent usage and consumer products (von Schneidemesser
et al., 2023; Lewis et al., 2020; Read et al., 2012).

In light of the substantial shifts observed in real-world VOC emission profiles, our objective is to assess the extent to which
current emission inventories accurately reflect these changes. Furthermore, given the significant advancements in the physical
and chemical formulation of the EMEP model since the last evaluative studies on VOCs that were conducted in the 1990s
(Hov et al., 1997; Solberg et al., 2001), it is important to update our understanding of the model's current performance and
identify the factors influencing it. Therefore, the aims of this study are (a) to augment the VOC species set in the EMEP model
with tracers for individual VOC compounds, (b) conduct a comprehensive comparison of some key VOCs between the EMEP
model and ambient measurements, and (c) to employ the model in assessing the 'goodness' of speciated emissions, providing
insights into their quality and impact. For these purposes, we deployed a 'tracer' method, which allows us to input explicit
emissions into the model and compute concentrations of individual VOCs. This tracer method has been used for whole-year
comparisons in 2018 and 2019, and for comparisons during the 2022 EMEP intensive measurement period (IMP), as presented
in Sect. 4. The methodology is described in Sect. 3.

It is important to note that this work mainly concentrates on VOCs with simpler structures and shorter chains. This focus is
due to the greater availability of measurement datasets and emission estimates that underpin the model's parameterisation and
evaluation for these compounds, in comparison to VOCs with more complex structures, such as longer-chain hydrocarbons
(e.g., greater than C7), acids, and esters. As more data becomes accessible, particularly regarding boundary conditions and
90 sector-specific emission profiles, we intend to refine and update the model configuration in the future.

## 3 Methods

### 3.1 The EMEP MSC-W model

The EMEP MSC-W atmospheric chemistry transport model has been developed by the European Monitoring and Evaluation
Programme Meteorological Synthesizing Centre – West. It is an open-source Eulerian grid model used for applications ranging
from scientific research to policy development (Simpson et al., 2012, 2020; Jonson et al., 2006, 2017; Ge et al., 2021, 2023b;
van Caspel et al., 2023). In the default setup for European simulations as used here, the model uses 20 terrain-following vertical
layers, with the pressure ranging from around 1000 hPa (surface level) to 100 hPa (highest level). The lowest layer has a height
of about 50 m. The model output of surface concentrations are adjusted to be equivalent to 3 m above the surface as described



in Simpson et al. (2012). In this study, we utilise the most recent EMEP MSC-W model version rv5 (Simpson et al., 2023),
which features a significantly revised photolysis scheme (Cloud-$J$) compared to previous EMEP versions. As described in van
Caspel et al. (2023), the Cloud-$J$ implementation was specifically developed to include each of the photolysis reaction rates
($J$-values) present in the CRIv2R5Em chemical mechanism described below.

## 3.2  Chemistry mechanisms

Two chemistry mechanisms, CRIv2R5Em and EmChem19rc, have been utilised to develop VOC tracers and to investigate the
difference in model performance across different mechanisms. The CRIv2R5Em chemical mechanism is an EMEP adaptation
(Bergström et al., 2022) of the Common Representative Intermediates (CRI) v2-R5 mechanism (Watson et al., 2008). This
mechanism is the simplest variant of CRI v2, considered suitable as a reference mechanism in large-scale chemistry-transport
models. The CRIv2R5Em also includes a recently developed isoprene reaction scheme (CRI v2.2a) that describes updates to
the major HOx recycling routes (Jenkin et al., 2019). A selection of 24 anthropogenic and 3 biogenic species are chosen to
represent all NMVOCs emitted in CRIv2R5Em, based on their Photochemical Ozone Creation Potentials (POCP, e.g. Jenkin
et al. 2017), abundance, and simplicity of mechanism. This EMEP adaptation (derived from a version based on CRIv2.1),
CRIv2R5Em, was created prior to the release of the latest CRI v2.2, hence it slightly differs from the current official CRIv2R5
version.

The EmChem19rc mechanism is a small update (Simpson et al., 2023) of EmChem19a. Both EmChem19a and CRIv2R5Em
are described in detail in Bergström et al. (2022). EmChem19rc is now the default chemical mechanism used in v5.0 of the
EMEP model. It typically employs primary emissions from 17 NMVOC species and surrogates (14 anthropogenic and 3
biogenic) to represent a wide variety of VOCs that are actually emitted into the atmosphere (Bergström et al., 2022). For
instance, n-butane (model species nC4H10) is utilised to represent both itself and other alkanes that contain more than three
carbon atoms, alongside a handful of other species with similar POCP. Similarly, benzene and toluene are explicit aromatic
VOC species, but then o-xylene is used as a surrogate for itself and all other aromatic VOCs having more than seven carbon
atoms. A detailed species list is presented in Supplement A, Table A1.

In order to obtain VOC concentrations from the model that are directly comparable with measurements, without affecting
computational efficiency and the mechanism's innate capability for ozone production, we employ a 'tracer' method. This
method retains the normal species set of CRIv2R5Em and EmChem19rc mechanisms for the calculation of photochemistry
(and hence OH, $O_3$ and $NO_3$ radical concentrations), but additionally introduces individual VOC tracers (denoted by the
suffix "_T") that take explicit emissions from a certain species and follow species-specific loss processes to yield precise
concentrations of that species. These tracers neither consume any atmospheric oxidants, like the OH radical, nor generate any
products; they are created solely to track VOC concentrations. For example, although emissions of i-butane ($iC_4H_{10}$) are lumped
with those of other heavy alkanes into the surrogate $nC_4H_{10}$ species for the standard photochemical calculations, we also track
the emissions and losses (using explicit OH + $iC_4H_{10}$ reaction rates) for the tracer species iC4H10_T. This procedure should
give the best estimate of its concentrations, assuming that the standard CRIv2R5Em and EmChem19rc model concentrations
of OH are reasonable – something which was demonstrated by Bergström et al. (2022).





Table 1 summarises available VOC species in the adapted CRIv2R5Em mechanism. Based on chemical species and reactions in CRIv2.2, several new species (coloured in blue in Table 1) are added to CRIv2R5Em as VOC tracers, which not only enable a comparison with EmChem19rc, but also with more measurements. Alongside these new species, additional tracers have also been created for existing lumped surrogates such as NC4H10_T, OXYL_T, and others (coloured in green in Table 1).

**Table 1.** Summary of current primary VOC species in CRIv2R5Em. Species coloured in blue are newly added VOC tracers; green indicates tracers for existing lumped surrogates; orange indicates species that have secondary production from lumped surrogates. TBUT2ENE represents 2-butene; NPROPOL and IPROPOL represent 1-propanol and 2-propanol respectively; GLYOX and MGLYOX represent glyoxal and methylglyoxal respectively; MEK represents methyl ethyl ketone. XTERPENE is a lumped surrogate for other biogenic species.

| CRIv2R5Em | Species | | | |
|---|---|---|---|---|
| Shorter-chain alkane | C2H6_T | C3H8 | NC4H10_T | IC4H10_T |
| Longer-chain alkane | NC5H12_T | IC5H12_T | NC6H14_T | NC7H16_T |
| Alkene | C2H4_T | C3H6_T | TBUT2ENE | |
| Alkyne | C2H2 | | | |
| Aromatics | BENZENE | TOLUENE | OXYL_T | |
| Alcohol | CH3OH | C2H5OH_T | NPROPOL | IPROPOL |
| Aldehyde | HCHO | CH3CHO | | |
| Dialdehyde | GLYOX | MGLYOX | | |
| Ketone | CH3COCH3 | MEK | | |
| Carboxylic acid | HCOOH | CH3CO2H | | |
| Biogenic VOC | C5H8 | $\alpha$-PINENE | $\beta$-PINENE | XTERPENE |
| Rest[†] | OTH_ALKANE_T | | | |

Notes †: Rest includes other alkanes and some other species.

## 3.3 Emissions

### 3.3.1 Current challenges

Ideally we hope to use individual species, or ratios of species, to identify particular emission sources. For example, Peischl et al. (2013) suggested that ethane emissions are dominated by natural gas supply infrastructure, whilst ethene, propene, and ethyne mainly originate from tailpipe exhaust (Coggon et al., 2021). However, concentrations of VOCs at measurement stations, often situated away from urban or industrial emission sources, are influenced by multiple contributors. VOCs with lifetimes of several days or longer become mixed with emissions from various sources by the time they are detected at the background stations. For instance, aromatic species such as toluene are emitted from both vehicular transport (Gkatzelis et al., 2021) and solvent usage (Mo et al., 2021). Similarly, butanes are released from fossil fuel usage but are also commonly used as aerosol propellants in various chemical products (Lewis et al., 2020).



Another significant challenge in creating speciated emissions for atmospheric models arises from the grouping of emissions in inventories. For instance, methanol, which has substantial biogenic emissions, is incorporated into several inventories (Guenther et al., 2012) and atmospheric models. Nevertheless, the representation of anthropogenic emissions in databases such as EDGAR (Huang et al., 2017) or CAMS (Kuenen et al., 2022) is more generic, listing alcohols as a collective category without specifying the proportions of methanol, ethanol, or other alcohols. This aggregation in global and regional anthropogenic VOC (AVOC) inventories no longer aligns with the advanced chemical mechanisms now employed in atmospheric models.

### 3.3.2 Emissions in the EMEP model

To address these challenges, we utilised the UK National Atmospheric Emission Inventory (UK NAEI), provided by the NAEI team upon email request in 2022, as the primary source of AVOC emission profiles for the work presented here. The key advantage of this inventory is its extensive coverage: it offers emissions data for 664 VOC species across 249 sectors, spanning the period from 1990 to 2019. Despite being based on a somewhat dated speciation profile developed in the early 2000s (Passant, 2002), this inventory is highly valuable. Given the paucity of national speciated VOC emission inventories reported by other European countries, the UK NAEI remains a valuable reference source.

The VOC emissions in the EMEP model consist of biogenic VOC (BVOC), which are calculated online from temperature, radiation and land-cover data (Simpson et al., 1999, 2012), biomass-burning emissions from the Fire INventory from NCAR (FINN) v2.5 (Wiedinmyer et al., 2023), and gridded AVOC inventories provided by the EMEP Centre for Emission Inventories and Projections (CEIP, www.emep.int), or through the Copernicus Atmosphere Monitoring Service (CAMS) projects (Kuenen et al., 2022; Denier van der Gon et al., 2023). The AVOC emissions (here we used the dataset CAMS-REG-v5.1.) are provided as sector-specific totals (e.g. VOC from solvents or road traffic sectors), and are the main focus of this study.

Monthly (and also day-to-day and hourly) time-factors are specified in the model as described in Simpson et al. (2023). Briefly, these time-factors (CAMS_TEMPO_CLIM in EMEP notation) correspond to the CAMS-REG-TEMPO v3.2 simplified climatological temporal profiles (Guevara et al., 2021; Guevara, 2023), but are updated for non-livestock agricultural emissions (GNFR Sector L) from CAMS-REG-TEMPO v4.1. Fig. B1 and B2 illustrate these monthly factors for two countries, the UK and Switzerland.

### 3.3.3 Emissions sector mapping

Emission sectors in the UK NAEI are mapped to the 19 EMEP sectors as shown in Table B1. The CAMS inventory reports emissions from sector-A (Public Power, abbreviated to PP) through A1 (PP-Point) and A2 (PP-Area), and emissions from sector-F (Road Transport) through F1 (Gasoline), F2 (Diesel), F3 (LPG), F4 (Non-exhaust). In contrast, the CEIP inventory only reports sector totals from A and F without specifying the exact emissions from each sub-sector. Given that most activities in the UK NAEI are classified into various Nomenclature for Reporting (NFR) sectors (which refers to the format for the reporting of national data in accordance with the Convention on Long-Range Transboundary Air Pollution), this mapping is achieved using a cross-walk between NFR and gridded aggregated nomenclature for reporting (GNFR) sectors as detailed in





Matthews and Wankmüller (2021). Most sources in the UK NAEI have corresponding emission profiles, with the exception of
activities falling under GNFR sectors A1, A2, F3, K (Agriculture-Livestock) and L (Agriculture-Other).

The VOC speciation for A1 and A2 is set to be the same as the sector A since UK NAEI does not include emissions from
such sub-categories for sector A. For sector F3, the VOC speciation for LPG exhaust is derived from the EMEP/CORINAIR
Emission Inventory Guidebook[1].

For sector K, we make use of data supplied by the Netherlands Organisation for Applied Scientific Research (TNO, A. Viss-
chedijk and J. Kuenen, pers.comm., 2023). These data consisted of European-scale country-specific emissions for individual K
sub-sectors (e.g., cattle, sheep, etc.) and for 25 VOC species or groups (e.g., voc01 alcohols, voc02 ethane, and voc04 butanes,
consistent with those used by Huang et al. 2017). For the grouped emissions in TNO dataset, we have utilised the following
references to split emissions from a certain group (e.g., voc01 alcohols) into individual species (e.g., methanol, ethanol, etc.)
for each country. For activities relating to poultry, cattle, and pigs, emission profiles reported by the EEA emission inventory
guidebook Chapter 3.B Manure management[2] are used. As for sheep-related activities, speciated VOC emissions from Hobbs
et al. (2004) are used.

Sector L encompasses activities such as the application of animal manure to soils and field burning of agricultural residues,
that can have vastly different emission profiles. Therefore, as well as the above-mentioned references, emission profiles for
field burning of agricultural residues from Andreae (2019) are also used to assign TNO's grouped emissions from this specific
sector to separate VOCs. For other sectors that do not have detailed speciation profiles available (e.g., sewage applied to soils;
cultivated crops; farm-level agricultural operations including storage, handling and transport of agricultural products), TNO's
grouped emissions are directly mapped to lumped surrogates in the EMEP model. The primary issue is the lack of publicly
available and speciated emission data for these activities. Moreover, in case that there are areas that are not covered by TNO
emissions in the modelled domain, a default emission speciation based on the EEA emission inventory guidebook [3] and Hobbs
et al. (2004) has been established for sector K and L at present. We plan to review these speciations when relevant data becomes
available in the future.

### 3.3.4  Emissions species mapping

The original species mapping between the NAEI species and EMEP compounds was developed by Garry D. Hayman[4] as
part of the study that was eventually published as Bergström et al. (2022). This mapping was based on an older version of
the UK NAEI,  and used EmChem09 chemical compounds (Simpson et al., 2012) and SNAP emissions sectors. Updates to
EmChem19a were conducted by Bergström et al. (2022), and we have further updated the mapping for the VOC speciation
described here. Figure 1 illustrates the mapping process from the NAEI sectors ($NS$) and VOC ($NV$) emissions to EMEP
VOCs ($EV$) and EMEP sectors ($ES$). Utilising raw data from the NAEI for a selected year, the total emissions for EMEP

---

[1]https://www.eea.europa.eu/publications/EMEPCORINAIR5/B710vs6.0.pdf/view
[2]https://www.eea.europa.eu/publications/emep-eea-guidebook-2019/part-b-sectoral-guidance-chapters/4-agriculture/3-b-manure-management/view
[3]https://www.eea.europa.eu/publications/emep-eea-guidebook-2019/part-b-sectoral-guidance-chapters/4-agriculture/3-b-manure-management/view
[4]now at UK Centre for Ecology and Hydrology




sector $i$, denoted $ES_i$, are calculated as follows:

$$ES_i = \sum_{j=1}^{n} EV_{j,1} + EV_{j,2} + \cdots + EV_{j,44}$$

where $j$ is the corresponding NAEI sector, $n$ is the total number of NAEI sectors that belong to $i$, and $EV_{j,1}$ to $EV_{j,44}$ represent emitted masses of up to 44 EMEP speciess (VOC tracers + lumped surrogates). In EMEP sector $i$, the percentage of the EMEP VOC $x$, $P_{i,x}$, is calculated as follows:

$$P_{i,x} = \frac{\sum_{j=1}^{n} EV_{j,x}}{ES_i} \times 100\%$$

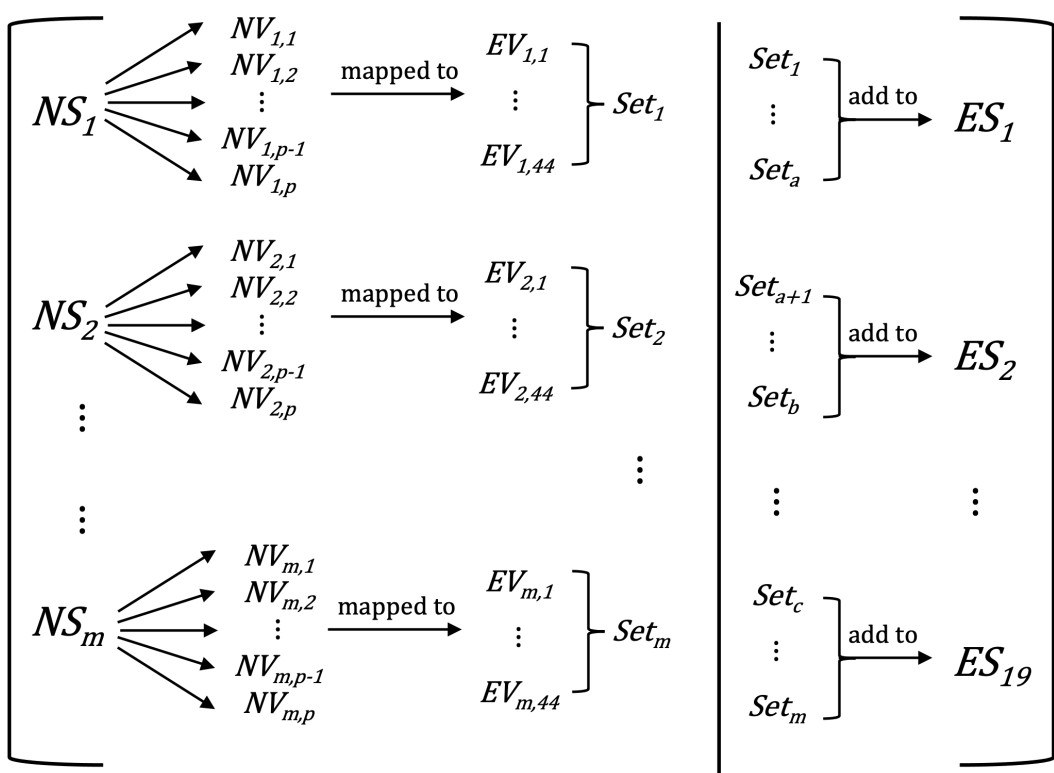

**Figure 1.** The emission mapping of NAEI VOCs ($NV$) from NAEI sectors ($NS$) to EMEP VOCs ($EV$) and EMEP sectors ($ES$). The total number of $NS$ is denoted by $m$; the total number of $NV$ is denoted by $p$.

Figure 2 presents the annual total VOC emissions for individual EMEP sectors in both inventories, as well as each sector's emission profiles implemented in the model (CRIv2R5Em). For the sector-K Agri-Livestock and sector-L Agri-Other, the country-specific speciation vary from one country to another, thus only the default spciation is shown. The CAMS inventory, with its smaller domain, generally reports lower sector totals than the CEIP inventory.

As described in Section 3.3.3, in the CAMS inventory, emissions from the sector-F Road Transport (RT) are reported in four sub-sectors (i.e., F1 = RT-Gasoline, F2 = RT-Diesel, F3 = RT-LPG, and F4 = RT-Non-exhaust), and thus its total is shown as the



sum of emissions from these sub-sectors. In contrast, the CEIP inventory only reports emissions from sector-F as a whole, and emissions for its sub-sectors are all set to zero (hence, emission profiles for these sub-sectors are not used in model simulations). The speciation profile of sector-F that is actually used for the CEIP inventory is derived from the individual speciation profile of F1, F2, F3, F4 and total emissions of these sub-sectors reported by the CAMS inventory. For a given EMEP VOC $x$, its

percentage in sector-F, $P_{F,x}$, is calculated as follows:

$$P_{F,x} = \frac{\sum_{i=1}^{4} ST_{Fi} \times P_{Fi,x}}{\sum_{i=1}^{4} ST_{Fi}} \times 100\%$$

where $ST_{Fi}$ represents the sector total emissions of each F sub-sector (F1, F2, F3, F4), and $P_{Fi,x}$ is the percentage of the EMEP VOC $x$ within the individual profile of each sub-sector

For the sector-A Public Power and its two sub-sectors, the same emission profile developed for PP using the NAEI data

is used for all these sectors. The reason why there are three sectors for PP is because different inventories report emissions in different formats. The CAMS inventory reports PP emissions in the format of PP-Point and PP-Area, whereas the CEIP inventory only reports emissions from the PP sector. Apart from these differences, both inventories indicate that the significant VOC emitting sectors include Fugitive, Solvents, and Road Transport.

Utilizing 2018 anthropogenic emissions data, more than 600 VOCs from the NAEI, in addition to several other VOCs from

the EEA emission inventory guidebook, are mapped to 44 EMEP species or groups. Figure 2 illustrates the 19 most substantially emitted species or groups, with less-emitted species and most lumped surrogates incorporated into the REST group.

It is worth noting that there are anthropogenic emissions of what are traditionally recognised as biogenic VOCs, and these are incorporated within the BVOC group in Fig. 2. This group represents anthropogenic emissions of isoprene, $\alpha$-pinene, $\beta$-pinene, and some terpene species. Anthropogenic emissions of the BVOC group are essentially only from the Industry sector and are

considerably smaller than their biogenic emissions (Borbon et al., 2023). The OTH_ALKANE group signifies emissions of higher alkanes, as well as some other complex VOCs. The UNREAC group represents emissions of species with low or no reactivity.



**Figure 2.** Annual total emissions (upper panel) from CAMS and CEIP inventories, and VOC profiles (lower panel) of individual EMEP sectors in CRIv2R5Em mechanism in 2018. Among the last 6 subsectors, PP stands for Public Power, RT stands for Road Transport. The speciation of sector F is an overall reflection of F1–F4. Note that CEIP do not provide data for the last six sectors (A1,A2,F1–F4), so emissions are zero for these sectors.




### 3.3.5 Speciation of biomass burning emissions

Table 2 displays the emission splitting factors used in the EMEP model for biomass burning species in the FINN inventory.
While FINN typically provides emissions data for individual species, it only offers a combined emission for butane species.
Consequently, the VOC speciation data derived from Andreae (2019) is employed to determine the ratios of n-butane to i-butane.

**Table 2.** The mapping between biomass burning species in the
FINN inventory and CRIv2R5Em species in the EMEP model.

| FINN species | model species | Factor |
| --- | --- | --- |
| C2H6 | C2H6_T | 1 |
| C3H8 | C3H8 | 1 |
| ALK4 | NC4H10_T | 0.6255 |
| ALK4 | IC4H10_T | 0.3745 |
| C2H4 | C2H4_T | 1 |
| C2H2 | C2H2 | 1 |
| PRPE | C3H6_T | 1 |
| XYLE | OXYL_T | 1 |
| BENZ | BENZENE | 1 |
| TOLU | TOLUENE | 1 |
| CH2O | HCHO | 1 |
| ALD2 | CH3CHO_T | 1 |
| GLYX | GLYOX | 1 |
| MGLY | MGLYOX | 1 |
| ACET | CH3COCH3 | 1 |
| MEK | MEK_T | 1 |

### 3.3.6 Boundary and initial conditions

The EMEP model specifies the boundary and initial conditions (BICs) of a number of compounds, including VOCs, using sim-
250 ple functions to describe changes from month to month, and accounting for altitude effects Simpson et al. (2012, 2015). BICs
are specified using a cosine function: $\chi_0 = \chi_{\text{mean}} + \Delta\chi \, \cos\left(2\pi \frac{(d_{\text{mm}} - d_{\text{max}})}{n_y}\right)$, where $\chi_0$ is the monthly near-surface concentra-
tion, $\chi_{\text{mean}}$ is the annual mean near-surface concentration, $\Delta\chi$ the amplitude of the cycle, $n_y$ is the number of days per year,
$d_{mm}$ is the day number of mid-month (assumed to be the 15th), and $d_{\text{max}}$ is day number at which $\chi_0$ maximises. Changes in
the vertical are specified with a scale-height, set to 10 km for C2H6 and 6 km for other VOC.
We endeavoured to base the BICs for the VOCs in this study predominantly on data from peer-reviewed literature. Table 3
shows the BICs used for the VOC compounds. The BICs for propane, n-butane, i-butane, n-pentane, and i-pentane are derived



from the average concentrations from a five-year dataset of high-frequency, in-situ VOC measurements taken at Mace Head, Ireland, as documented by Grant et al. (2011). A numerical factor is used to partition the boundary condition of the lumped species C4H10 into six VOC species or groups: C3H8, NC4H10_T, IC4H10_T, NC5H12_T, IC5H12_T, and OTH_ALKANE_T. For ethane and ethyne, their BICs are derived from ten-year average concentrations measured at three French rural background sites reported by Waked et al. (2016).

**Table 3.** The boundary and initial conditions (BICs) for VOC species in the EMEP model. See text for explanation of terms.

| model species | $\chi_{mean}$ (ppb) | $\Delta\chi_{mean}$ (ppb) | $d_{max}$ |
|---|---|---|---|
| C2H6_T | 1.544 | 0.77 | 75 |
| C2H2 | 0.456 | 0.23 | 75 |
| C3H8 | 0.263 | 0.13 | 45 |
| NC4H10_T | 0.095 | 0.05 | 45 |
| IC4H10_T | 0.044 | 0.02 | 45 |
| NC5H12_T | 0.026 | 0.01 | 45 |
| IC5H12_T | 0.026 | 0.01 | 45 |
| OTH_ALKANE_T | 1.546 | 0.77 | 45 |
| HCHO | 0.7 | 0.3 | 180 |

## 3.4 Measurements

The measurement data in the regular EMEP monitoring network are documented in the EMEP annual VOC reports (e.g. Solberg et al., 2020, 2022) and references therein. Detailed measurement guidelines such as analytical techniques, calibration procedures, and QA/QC measures are described in (Reimann et al., 2018). Measurement data used in this study are compiled from the Ebas platform (ebas-data.nilu.no), including both regular measurements for the year 2018 and 2019, and the 2022 EMEP Intensive Measurement Period (IMP) campaign. The IMP was organised by the EMEP Task Force on Measurement and Modelling (TFMM) in 12-19 July 2022. One-week observations of VOCs relevant as ozone precursors was conducted, covering both EMEP background sites and many urban sites. In this work, only some OVOC measurements from IMP are used as a supplement since the normal EMEP monitoring network only has one or two sites available for OVOC species. A separate IMP-focused paper is in preparation by other research teams, with the purpose of improving our current understanding of the formation of ozone during heat waves.

Table 4 presents a summary of the codes, names, and altitudes of all stations referenced in this study, for both the whole year of 2018 and 2019, and the 2022 IMP. The locations of these sites are shown in Fig. 3. Stations situated above 800 m in altitude are omitted from all analyses, to reduce problems associated with comparison with modelled surface concentrations.



**Table 4.** Codes, names, countries, and altitudes (m) of stations providing VOC measurements used in this study.

| Code | Name | Country | Altitude/m | Code | Name | Country | Altitude/m |
|------|------|---------|------------|------|------|---------|------------|
| AT0002R | Illmitz | Austria | 117 | FI0096G | Pallas | Finland | 565 |
| BE0007R | Vielsalm | Belgium | 496 | FR0013R | Peyrusse Vieille | France | 200 |
| CH0053R | Beromünster | Switzerland | 797 | FR0015R | La Tardière | France | 133 |
| CZ0003R | Kosetice | Czechia | 535 | GB0048R | Auchencorth Moss | UK | 260 |
| DE0002R | Waldhof | Germany | 74 | GB1055R | Chilbolton Observatory | UK | 78 |
| DE0007R | Neuglobsow | Germany | 62 | IE0031R | Mace Head | Ireland | 5 |
| DE0009R | Zingst | Germany | 1 | IT0004R | Ispra | Italy | 209 |
| ES0021U | Madrid | Spain | 669 | NO0002R | Birkenes II | Norway | 219 |
| FI0050R | Hyytiälä | Finland | 181 | | | | |

## 3.5 Model experiments

The meteorology data is generated using the European Centre for Medium-Range Weather Forecasts(ECMWF) model. The VOC tracers and their related code are integrated into the EMEP model via the GenChem system (Simpson et al., 2020). It utilises a chemical pre-processor GenChem.py to convert chemical equations into differential form and generate the corre-
sponding FORTRAN code for use in the EMEP model. Given the differences in emissions between CEIP and CAMS, especially for some key sectors (c.f. Fig. 2), we run simulations with both inventories, and also with both EmChem19rc and CRIv2R5Em.

For the model evaluation purpose, six model simulations (Table 5) were carried out at a grid resolution of $0.1° \times 0.1°$ over the Europe domain, covering the 2018, 2019 and 2022 periods. Additionally, although a full evaluation of the impacts of uncertainties in VOC speciation on ozone is beyond the scope of this study, we have set up two additional model runs for 2018
( (Table 5) ), Em-CAMS-2018-nDef (abbreviated to nDef hereinafter) and Em-CAMS-2018-Sol6 (Sol6), to output extra model variables for ozone and compared their differences between the two runs. The VOC speciation used in nDef is the same as the one used in the Em-CAMS-2018 model run, while in Sol6 the speciation of the sector E (Solvents) has been replaced by that of sector F1 (Road Transport-Gasoline). Detailed information on this sensitivity test is given at Sect. 4.6 and in the Supplementary Sect.G.





**Figure 3.** Locations of measurement sites providing VOC measurement used in this study

## 4 Results and discussions

This section provides a comparative analysis between modelled and measured surface VOC concentrations for the full years 2018 and 2019, using measurements from the standard EMEP monitoring network (Solberg et al., 2020), as well as from 2022 IMP. The comparison is complicated by the variation in the number of monitoring sites per species and in the frequency and duration of sampling time across stations. For example, the sampling duration for benzene varies from 5 to 40 minutes from DE0002R to GB0048R sites, while the model only calculates standard hourly concentrations. For this work we have matched the hourly model outputs with valid measurements at their native temporal resolution wherever we can. For instance, when using online Gas Chromatography (GC) measurements with an hourly resolution, such as CH0053R, we utilise the standard





**Table 5.** Configuration of model simulations.

| Simulation | Mechanism | Emission |
|---|---|---|
| Em-CEIP-2018 | EmChem19rc | CEIP |
| Em-CAMS-2018 | EmChem19rc | CAMS-REG-v5.1 |
| CRI-CEIP-2018 | CRIv2R5Em | CEIP |
| CRI-CAMS-2018 | CRIv2R5Em | CAMS-REG-v5.1 |
| CRI-CAMS-2019 | CRIv2R5Em | CAMS-REG-v5.1 |
| CRI-CAMS-2022 IMP | CRIv2R5Em | CAMS-REG-v5.1 |
| Em-CAMS-2018-nDef | EmChem19rc | CAMS-REG-v5.1 |
| Em-CAMS-2018-Sol6 | EmChem19rc | CAMS-REG-v5.1 |

hourly model outputs. In contrast, for VOC measurements collected using the steel canister method (for example, FR0013R), these are compared with four-hour model averages (spanning 12:00 to 16:00) on the sampling day. This time frame is commonly

used for canister sampling analysis, and the precise timing and duration of sampling within this time window often vary from one station to another. Therefore, due to the challenge in ascertaining these operational specifics for each station and species, we employ a model average over this period for comparison with the measured concentrations. Moreover, the annual mean concentrations discussed in this section are derived from hours with valid measurements, and where the sites have at least 65% data capture in a year.

The evaluation was undertaken for all model experiments. The comparisons for 2018 and 2019 show similar characteristics, and simulations with the different mechanisms were also similar. To avoid repetition, figures in this section are derived from the model simulation utilising the CRIv2R5Em mechanism and the 2018 CAMS inventory, unless otherwise specified.

## 4.1 Alkane species

### 4.1.1 Shorter-chain alkanes

Scatter plots comparing the modelled and measured annual mean concentrations of four shorter-chain alkane species, ethane, propane, n-butane, and i-butane (also known as 2-methylpropane) from the CRI-CAMS model runs are depicted in Figs. 4 and 5 for 2018 and 2019, respectively. A summary of the evaluation statistics for each model simulation is provided in Table 6.

The model and measurements agree that ethane has the highest annual concentrations of these alkanes, at around 1.7 ppb, followed by propane and n-butane. While the model-measurement comparisons in 2019 demonstrate slightly improved linear

correlation coefficients relative to those in 2018 as shown in Figs. 4 and 5, modelled concentrations of these species at both years show consistent underpredictions for propane (up to -54%) and i-butane (up to -38%) but overpredictions (up to +55%) for n-butane as shown in Table 6.

Issues with boundary conditions could partially account for the significant underestimation by the model, particularly concerning propane. Noticeable spatial variations in propane concentrations at background stations are observed in several studies.





Grant et al. (2011) reported average concentrations of propane over a 2005-2009 period at the representative Northern Hemi-sphere background station of Mace Head of 263 ppt in baseline air and 452 ppt in European transported air masses. Sauvage et al. (2009) showed that multi-year average propane levels in early 2000s varied in the range 576-731 ppt among three French rural sites. Dollard et al. (2007) reported an annual mean value of 832 ppt at rural UK sites in 2000. Consequently, the BICs used in our model, which is derived from five-year averages in the clean baseline air masses as reported by Grant et al. (2011),

may not effectively capture such spatial fluctuations.

Another contributing explanation may simply be an underprediction of propane emissions. Dalsøren et al. (2018) found much better agreement between modelled and observed propane when updated emissions from both natural and anthropogenic sources were included in place of their base CEDS emissions (Hoesly et al., 2018), and emissions of propane due to leakage from pipelines and other sources is hard to estimate reliably; it is not clear if the European EMEP emissions suffer from similar

underestimates.

The simulations in 2018 using the four model setups produce very similar statistical results for each alkane (Table 6). Ranking the model performances between different model simulations, those utilizing the CAMS inventory display slightly better comparison results than those utilizing the CEIP inventory. Possible reasons for improvement include the inclusion of more detail in the road traffic emissions sectors (F1–F4) in CAMS, and differences in absolute amounts and spatial distributions

of the emissions, but the modelled results for alkanes are obviously not particularly sensitive to these differences.

### 4.1.2 VOC ratios: propane/ethane

The agreement between modelled and measured VOCs varies with species, even among VOCs that are commonly understood to originate from the same emission sources. Comparisons of ethane and propane are a good example. Ambient levels of ethane and propane are primarily influenced by leakage from the production and usage of oil and natural gas (von Schneidemesser

et al., 2010; Aydin et al., 2011; Malley et al., 2015). Ethane, with a relatively long lifetime of around 1 month, exhibits a relatively low sensitivity to local emissions and is therefore an ideal species for the evaluation of a regional model at 0.1 degree. In contrast, spatial concentrations of propane, which has a lifetime approximately one-fourth that of ethane (c.f. Sect. C1), are more sensitive to local emissions (Plass-Dülmer et al., 2002; Franco et al., 2015; Helmig et al., 2016).

Our time series comparisons reveal a consistent temporal pattern between the model and measurement for ethane (Fig. D1),

but a larger discrepancy between modelled and measured propane concentrations (Fig. D2), particularly during winter and early spring months when modelled concentrations are considerably smaller than measurement peaks. Considering that both species have well-constrained OH loss rates and their rate coefficients have similar temperature dependence (Jenkin et al., 1997, 2008; Saunders et al., 2003; Watson et al., 2008), it is highly unlikely that the kinetics for these very straightforward reactions are not described properly in the chemical mechanism. More importantly, given the increased usage of fossil fuels in winter for e.g.

domestic heating and road transport purposes, this discrepancy likely indicates either missing propane emissions from sources like natural gas, or an underestimation of total sector emissions from the LPG sector, where propane emissions are predominant (see Fig. 2) — or possibly both.







**Figure 4.** Scatter plots of annual mean modelled and measured shorter-chain alkanes concentrations in 2018. The term 'CRI' indicates that the model data is calculated using the CRIv2R5Em mechanism. In each plot, the grey line is the 1:1 line, and the other coloured line is the least-squares regression line.

Additionally, the GB1055R site (Chilbolton Observatory), while set up as a rural background site, exhibits a year-round pattern of pronounced spikes in measured propane concentrations, in contrast to modelled values. These spikes, often exceeding





**Figure 5.** Scatter plots of annual mean modelled and measured shorter-chain alkanes concentrations in 2019. The term 'CRI' indicates that the model data is calculated using the CRIv2R5Em mechanism. In each plot, the grey line is the 1:1 line, and the other coloured line is the least-squares regression line.

10 ppb—while peak concentrations at other sites remain below 2 ppb—suggest that there are strong local sources nearby or



**Table 6.** Summary of the comparison statistics between the model ($M$) and observation ($O$) for shorter-chain alkane species. $N$ is the number of sites. $R$ is the Pearson's correlation coefficient between annual means at various sites. Mean_O and Mean_M refer to the annual average concentrations (in ppb) of $O$ and $M$ over all sites, respectively. NMB is the Normalised Mean Bias, and NME is the Normalised Mean Error.

| C2H6_T | N | R | Mean_O | Mean_M | NMB | NME |
|---|---|---|---|---|---|---|
| Em-CEIP-2018 | 10 | 0.5400 | 1.695 | 1.478 | -13% | 13% |
| Em-CAMS-2018 | 10 | 0.5884 | 1.695 | 1.463 | -14% | 14% |
| CRI-CEIP-2018 | 10 | 0.5318 | 1.695 | 1.484 | -12% | 13% |
| CRI-CAMS-2018 | 10 | 0.5781 | 1.695 | 1.469 | -13% | 13% |
| CRI-CAMS-2019 | 9 | 0.6988 | 1.709 | 1.489 | -13% | 14% |
| **C3H8_T** | **N** | **R** | **Mean_O** | **Mean_M** | **NMB** | **NME** |
| Em-CEIP-2018 | 10 | 0.3343 | 0.660 | 0.330 | -50% | 50% |
| Em-CAMS-2018 | 10 | 0.4858 | 0.660 | 0.292 | -56% | 56% |
| CRI-CEIP-2018 | 10 | 0.3263 | 0.660 | 0.335 | -49% | 49% |
| CRI-CAMS-2018 | 10 | 0.4748 | 0.660 | 0.296 | -55% | 55% |
| CRI-CAMS-2019 | 9 | 0.6725 | 0.678 | 0.310 | -54% | 54% |
| **NC4H10_T** | **N** | **R** | **Mean_O** | **Mean_M** | **NMB** | **NME** |
| Em-CEIP-2018 | 9 | 0.6003 | 0.245 | 0.373 | 52% | 60% |
| Em-CAMS-2018 | 9 | 0.6130 | 0.245 | 0.361 | 47% | 56% |
| CRI-CEIP-2018 | 9 | 0.5963 | 0.245 | 0.381 | 55% | 62% |
| CRI-CAMS-2018 | 9 | 0.6093 | 0.245 | 0.368 | 50% | 59% |
| CRI-CAMS-2019 | 9 | 0.6087 | 0.266 | 0.386 | 45% | 50% |
| **IC4H10_T** | **N** | **R** | **Mean_O** | **Mean_M** | **NMB** | **NME** |
| Em-CEIP-2018 | 9 | 0.3935 | 0.148 | 0.097 | -35% | 41% |
| Em-CAMS-2018 | 9 | 0.4038 | 0.148 | 0.093 | -38% | 42% |
| CRI-CEIP-2018 | 9 | 0.3974 | 0.148 | 0.099 | -34% | 40% |
| CRI-CAMS-2018 | 9 | 0.4083 | 0.148 | 0.095 | -36% | 41% |
| CRI-CAMS-2019 | 9 | 0.5201 | 0.148 | 0.103 | -30% | 30% |

pollution plumes regularly passing over possibly originating from urban areas such as London. This pattern further underscores the significant impact of local emissions on ambient propane levels.

Figure 6 illustrates a strong correlation between ethane and propane for both modelled and measured concentrations at two representative sites from Switzerland and Germany, which indicates common sources for the two species in all seasons. The measured propane-to-ethane ratios vary around 0.5, whereas the modelled ratios are around half of that value. These





modelled ratios display even lower values during the spring months compared to those in autumn, which aligns with the aforementioned model underestimation of propane concentrations in early spring. This discrepancy underlines the necessity for more investigations into the seasonal variations in emission profiles and sector-specific contributions in the inventory to more accurately reflect real-world ratios of the two species.

Figure 6. Measured (left) and modelled (right) propane to ethane ratios at Switzerland and German sites in 2018. Spring: Mar-May; Summer: Jun-Aug; Autumn: Sep-Nov; Winter: Dec, Jan-Feb





### 4.1.3 VOC ratios: i-/n-butane

Our study also highlights issues concerning the ratios among VOC isomers such as i-butane and n-butane. Several studies have reported that the i-/n-butane ratio has remained relatively constant at around 0.6 in recent decades (Helmig et al., 2014; Zhang et al., 2013; Parrish et al., 1998). However, as discussed in Sect. 4.1.1, the model tends to overestimate n-butane concentrations whilst underestimating those of i-butane. Consequently, the model simulates lower i-/n-butane ratios than those observed in measurements. As the two isomers have rather similar chemical loss rates (with lifetimes of ca. 3-4 days at [OH] $= 1.5 \times 10^6$ molec·cm$^{-3}$, c.f. Tab. C1, or 1 day at more typical summertimef [OH] $= 5.0 \times 10^6$ molec·cm$^{-3}$), one would expect good correlation between the two. Figure 7 shows that strong linear correlations are indeed observed between individual measured i- and n-butane samples at two example sites in Switzerland and the UK. The measured i-/n-butane ratios are approximately 0.6 and are similar across different seasons, implying that both isomers are likely to originate from the same sources throughout the year but with differing emission strengths. In contrast, although the model simulates strong linear correlations between the two isomers at both sites, the ratio is notably lower, ranging between 0.20 and 0.23. The modelled ratios are consistently lower than the observed ratios at other sites also.

From Table 2 we can calculate an i- to n-butane ratio in biomass burning emissions of 0.6, so wildfires episodes would not lower the ratio. Considering that there is no substantial difference in the atmospheric lifetimes of the two isomers, the smaller modelled ratios must be attributable to smaller ratios in the anthropogenic emissions. The overall emission ratios using the CAMS inventory data, as shown in Fig. D3, differ significantly among countries, ranging from around 0.1 in western Europe to around 0.4 in northern Europe. Examining individual emission sectors, Figure 8 identifies the solvent sector as the largest contributor to n-butane emissions, with an i-/n-butane ratio below 0.05. This is followed by the fugitive sector, which has a ratio of approximately 0.28. Consequently, a model grid with a larger proportion of emissions from the solvent and fugitive sectors would exhibit smaller ratios. In contrast, a greater contribution from biomass burning emissions or the RT-Non-exhaust sector (which has a ratio of 0.75) in a model grid on a specific day would yield larger ratios. This explains why in Fig. 7 certain model data points in winter exhibit substantially lower ratios compared to those in spring. Indeed, Fig. B1 and B2 further confirms that while solvent sector does not show large seasonal variations in Switzerland and the UK, VOC emissions from the other combustion and fugitive sectors are higher in winter (both have low ratios), which is consistent with the lower modelled ratios in winter at the two sites.

Our results also align with UK-AQEG (2020) who reported that the contribution of solvents to UK emissions was approximately 74% in 2017. Moreover, n-butane — primarily originating from solvents and fugitive losses — was as the second most abundant VOC by mass in the UK NAEI inventory, after ethanol. However, the divergence in modelled and measured i-/n-butane ratios revealed in this work suggests that this may not be the case in reality. Several studies have highlighted considerable discrepancies between emission inventories and ambient observations, particularly in relation to the dominant sources of VOC emissions (Niedojadlo et al., 2007; Lanz et al., 2008; Gaimoz et al., 2011). For instance, Borbon et al. (2013) suggested that emissions from gasoline-powered vehicles continue to be the dominant source of NMHC in northern mid-latitude urban areas, whereas several European regional inventories have identified solvent usage as the new leading urban VOC source.





Oliveira et al. (2023) also noted that estimates of solvent speciation differed considerably between different sources. Conse-

400 quently, it is possible that emissions of i-butane from the solvent sector may be unaccounted for, or that contributions from road transport-related sectors are underestimated, or both. Hence, further examination is required both of the sector that dominates emissions and of the specific emission profiles within each sector.

**Figure 7.** Measured (left) and modelled (right) i-butane to n-butane ratios at Switzerland and UK sites in 2018. Spring: Mar-May; Summer: Jun-Aug; Autumn: Sep-Nov; Winter: Dec, Jan-Feb







**Figure 8.** Annual total emissions of i-butane and n-butane from individual sectors in the 2018 CAMS inventory, and their emitted ratios.



### 4.1.4 Longer-chain alkanes

Figure 9 compares the modelled and measured annual mean concentrations of longer-chain alkane species — n-pentane, i-
pentane, and n-hexane — for the CRI-CAMS-2018 model run. These species exhibit generally lower concentrations than
shorter-chain alkanes with most sites reporting annual averages below 0.2 ppb. Despite the low concentrations, there is a good
agreement between the model and measurement for all species, with linear correlation coefficients ranging from 0.53 to 0.93.

Table 7 summaries the model-measurement comparison statistics of these species for each model simulation. In general,
simulations employing the CAMS inventory, which benefits from more detailed emissions data within CAMS's sub-sectors,
demonstrate stronger linear correlations compared to those utilising the CEIP inventory. No distinct differences are apparent
between simulations employing different chemical mechanisms. Similarly, the model's performance for all species remains
consistently good for both the 2018 and 2019 simulations.

Additionally, it is worth noting that although i-pentane contributes a notable amount of VOC emissions, comparable to
those of n-pentane within sectors such as Fugitive, RT-Gasoline, and RT-Non exhaust (Fig. 2), the modelled concentrations
of i-pentane are not as overestimated as those for n-pentane (+44%). On the contrary, the model significantly underestimates
i-pentane concentrations, by as much as -59% (Table. 7). This discrepancy necessitates further investigation to determine
whether it stems from inaccuracies in the speciation profiles of existing emission activities or whether it highlights the lack of
representation of another source. Coll et al. (2010) reported similar issues with the underestimation of i-pentane and highlighted
its significance as a component of gasoline evaporation. Several studies suggest that this aspect is not adequately captured in
emission inventories, albeit it is expected to account for a significant proportion of emissions within urban environments
(Borbon et al., 2002; Möllmann-Coers et al., 2002). An analysis of the ratios of i- to n-pentane is presented in the subsequent
section (Sect. 4.1.5).

### 4.1.5 VOC ratios: i-/n-pentane

The dominant anthropogenic sources of pentane species are traffic exhaust and fuel evaporation (Wilde et al., 2021; Li et al.,
2017; Gilman et al., 2013; Swarthout et al., 2013). Helmig et al. (2014) noted that measurements influenced by anthropogenic
emission sources displayed i-/n-pentane ratios ranging from 1.8 to 2.5. Bourtsoukidis et al. (2019) reported an i-/n-pentane ratio
of 1.7 above the Suez Canal, which is indicative of ship emissions, and a ratio of 2.9 in areas under the influence of considerable
vehicle emissions. In contrast, sources such as biomass burning and oceanic emissions preferentially emit n-pentane, resulting
in lower i-/n-pentane ratios of around 0.5 to 0.7 (Andreae, 2019; Lewis et al., 2001; Broadgate et al., 1997). In the absence of
parameterisation data for biomass burning and oceanic emissions of pentane species at the time of our model experiments, the
model results are primarily determined by anthropogenic emissions.

Considering that the two pentane isomers are commonly co-emitted and have rather similar atmospheric lifetimes (ca. 2
days) (Table C1), their concentration ratios are relatively indicative of their overall emission ratios. Figure 10 reveals that
although strong linear correlations are present in both the modelled and measured datasets across all seasons, the measured
i-/n-pentane ratios (around 1.45) are more than three times higher than the modelled ratios (around 0.39). As with the model's





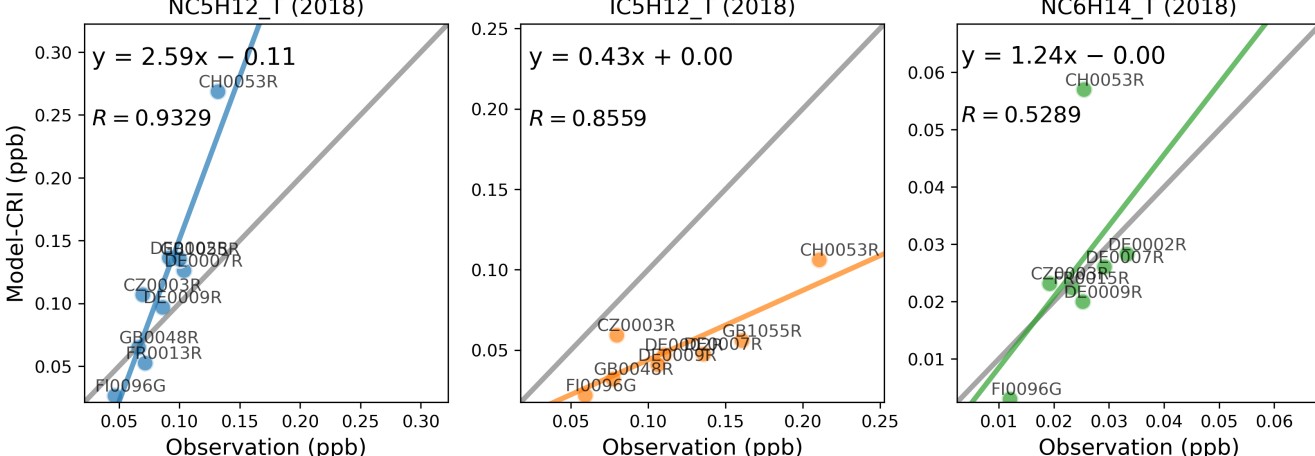

**Figure 9.** Scatter plots of annual mean modelled and measured longer-chain alkanes concentrations in 2018. The term 'CRI' indicates that the model data is calculated using the CRIv2R5Em mechanism. In each plot, the grey line is the 1:1 line, and the other coloured line is the least-squares regression line.

underestimation of i-/n-butane ratios, this smaller modelled i-/n-pentane ratio is largely driven by abundant n-pentane emissions from the solvent sector, which exhibits an exceedingly low emission ratio of only 0.003, as indicated in Fig. 11.

Besides potential speciation inaccuracies within the solvent sector, an alternative explanation for the discrepancy between modelled and measured ratios could be that current inventories underestimate total emissions from transport activities and fuel evaporation. In this case, the likelihood of the first hypothesis is supported by measurement data, which exhibit a strong linear correlation between the two species, with data points tightly clustered around the regression line and a slope representing one dominant emitting ratio, rather than displaying multiple trend lines with varying slopes. However, pinpointing which of the two possibilities accounts for the model's underestimation of i-/n-pentane ratios remains a challenge.

Additionally, it is pertinent to note that the current speciation for agricultural sectors, derived from available literature (Sect. 3.3.3), only contains n-pentane and not i-pentane, thereby contributing to the lower i-/n-pentane ratio calculated in the model. In fact, there is a general lack of emission measurements to support a detailed and accurate speciation of VOC emissions from agricultural sectors. These sectors contains a variety of activities, each with potentially different emission profiles and uncertainties. For instance, the Agri-Livestock sector comprises emissions from diverse animal categories such as poultry, cattle, sheep, and swine. Similarly, the Agri-Other sector includes activities ranging from the application of animal manure and the cultivation of crops to the field burning of agricultural residues. Figure 11 suggests that the contributions of emissions from agricultural sectors are not insignificant. As such, there is a pressing need for more emission measurements to enhance the accuracy of VOC emission speciation in these sectors.



**Table 7.** Summary of the comparison statistics between the model ($M$) and observation ($O$) for longer-chain alkane species. $N$ is the number of sites. $R$ is the Pearson's correlation coefficient between annual means at various sites. Mean_O and Mean_M refer to the annual average concentrations (in ppb) of $O$ and $M$ over all sites, respectively. NMB is the Normalised Mean Bias, and NME is the Normalised Mean Error.

| NC5H12_T | N | R | Mean_O | Mean_M | NMB | NME |
|---|---|---|---|---|---|---|
| Em-CEIP-2018 | 9 | 0.9100 | 0.085 | 0.107 | 26% | 38% |
| Em-CAMS-2018 | 9 | 0.9331 | 0.085 | 0.109 | 28% | 40% |
| CRI-CEIP-2018 | 9 | 0.9105 | 0.085 | 0.110 | 30% | 42% |
| CRI-CAMS-2018 | 9 | 0.9329 | 0.085 | 0.113 | 32% | 43% |
| CRI-CAMS-2019 | 9 | 0.8775 | 0.081 | 0.116 | 44% | 50% |
| **IC5H12_T** | **N** | **R** | **Mean_O** | **Mean_M** | **NMB** | **NME** |
| Em-CEIP-2018 | 8 | 0.7422 | 0.117 | 0.048 | -59% | 59% |
| Em-CAMS-2018 | 8 | 0.8592 | 0.117 | 0.050 | -58% | 58% |
| CRI-CEIP-2018 | 8 | 0.7382 | 0.117 | 0.050 | -58% | 58% |
| CRI-CAMS-2018 | 8 | 0.8559 | 0.117 | 0.051 | -56% | 56% |
| CRI-CAMS-2019 | 6 | 0.8866 | 0.107 | 0.050 | -53% | 53% |
| **NC6H14_T** | **N** | **R** | **Mean_O** | **Mean_M** | **NMB** | **NME** |
| Em-CEIP-2018 | 7 | 0.4428 | 0.024 | 0.022 | -8% | 44% |
| Em-CAMS-2018 | 7 | 0.5251 | 0.024 | 0.025 | 3% | 36% |
| CRI-CEIP-2018 | 7 | 0.4479 | 0.024 | 0.023 | -4% | 43% |
| CRI-CAMS-2018 | 7 | 0.5289 | 0.024 | 0.026 | 7% | 35% |
| CRI-CAMS-2019 | 7 | 0.4984 | 0.023 | 0.025 | 9% | 42% |





**Figure 10.** Measured (left) and modelled (right) i-pentane to n-pentane ratios at Germany and UK sites in 2018. Spring: Mar-May; Summer: Jun-Aug; Autumn: Sep-Nov; Winter: Dec, Jan-Feb



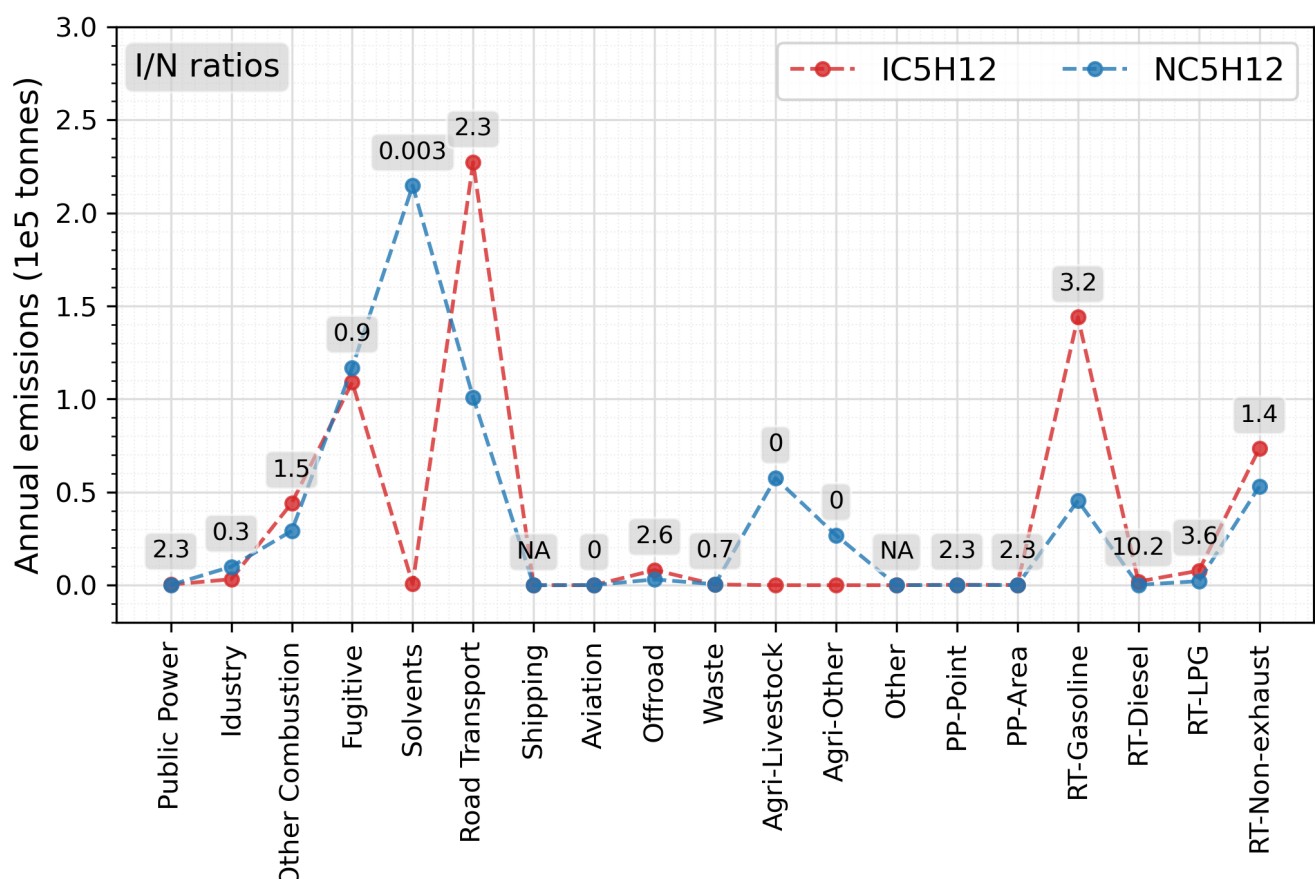

**Figure 11.** Annual total emissions of NC5H12 and IC5H12 from individual sectors and their emitted ratios using the CAMS inventory. 'NA' means there is no NC5H12 emissions





## 4.2 Unsaturated NMHCs

### 4.2.1 Ethene, ethyne and isoprene

Figure 12 presents a comparison of the annual mean concentrations of ethene, ethyne, and isoprene for the year 2018. The comparison statistics are presented in Table 8. Results are mixed, with rather good results for ethene, but poorer results for ethyne. As for isoprene, an outlier station is found for comparisons in both 2018 and 2019.

The statistical metrics for ethyne remained consistently poor in all 2018 simulations, in contrast to the much better performance for ethene. This is interesting given that both are emitted from similar anthropogenic activities and have well-constrained
OH loss rates (Jenkin et al., 1997, 2008; Saunders et al., 2003; Watson et al., 2008). The different model performance strongly points to shortcomings in the spatial and temporal patterns and magnitudes of ethyne emissions. Moreover, the model's poor performance for ethyne are twofold: it underestimates ethyne concentrations at most sites and fails to reflect the spatial distribution evident in the measurements. As illustrated in Figure 12, the dispersion of the data points along the measurement axis is considerable, indicating variability in the actual concentrations. In contrast, the model results tightly cluster around 0.35 ppb,
indicating little spatial variation. This clustering suggests that the modelled concentrations of ethyne are heavily influenced by its BICs, approximately 0.46 ppb, which are derived from ten-year average concentrations of measurements at three rural background sites in France reported by Waked et al. (2016). In our previous model runs in which similar anthropogenic emission profiles are used but the BICs and biomass burning emissions of ethyne had not been developed, the model underestimation was even larger at -91% and the linear correlation remained poor (R = -0.18) (Ge et al., 2023a). With the BICs and biomass
burning emissions for ethyne applied for model runs in this work, it appears that the inputs of anthropogenic emissions from both inventories are too small to significantly affect the model outputs.

A detailed time series comparison of ethyne at example stations (Fig. 13) reinforces this hypothesis. Compared to the measurement data, the model predicts lower concentrations of ethyne during the winter and early spring, with little seasonal fluctuations. Given that the model aligns relatively well with the observed low concentrations during the summer months, the
discrepancies likely stem from an underestimation of ethyne emissions from sources such as road transport activities or other combustion processes especially during winter.

Considering that ethyne is commonly used as a tracer of anthropogenic emissions, these discrepancies are important. von Schneidemesser et al. (2023) used ratios of VOC to ethyne, rather than VOC to CO to evaluate global inventories, because of measurement availability; but as they noted, the validity of such comparisons depend crucially on how well ethyne is
represented in the inventory. In fact, the observed peak concentrations of ethyne far exceed those we can model using the current inventories. A discussion on VOC to ethyne ratios is presented in Sect. 4.2.3 and 4.2.4 to delve into this issue in more detail.

For isoprene, it is one of the most important biogenic VOCs, whose emissions are dominated by its biogenic sources (Guenther et al., 2012; Simpson et al., 1999). Traffic related sources can also be important in wintertime (Reimann et al., 2000;
Borbon et al., 2001). Figure 12 and E1 show that an outlier station DE0007R, characterised by a large model overestimation, drives an overall very poor linear correlation coefficient (R = 0.22) in both 2018 and 2019. Figure E2 shows that both the




model and the measurement show peak concentrations in summer and very low concentrations in winter. The comparison is challenged by the model overestimation at site DE0007R, which is likely due to inaccuracies in simulating isoprene emissions from vegetation within this specific model grid. The biogenic emission field for isoprene is predominantly influenced by the

490 estimated locations of certain key species, such as the European oak, known for their high emission factors (Wiedinmyer et al., 2006). Consequently, the emission field can appear quite 'spotty'. Further investigation into the geographical characteristics of this site, both in the actual world and as represented in the model, indicates that it is situated in a forest hot spot in both scenarios. This leads to the possibility of a discrepancy in the exact type of forest depicted by the model compared to the actual one. Nevertheless, apart from this anomaly, the data points of other locations in both years show excellent linear correlations

between model and measurement, with R values equal to 0.92 in 2018 and 0.99 in 2019, despite a consistent model underestimation. The good agreements at those sites for this species are somewhat surprising due to both the difficulties associated with estimating the magnitude and spatial distribution of its biogenic sources (Simpson et al., 1999; Keenan et al., 2009) and its short lifetime, demonstrating that the model calculation of biogenic emissions for isoprene and its chemistry and transport is being captured reasonably well at least at these locations.

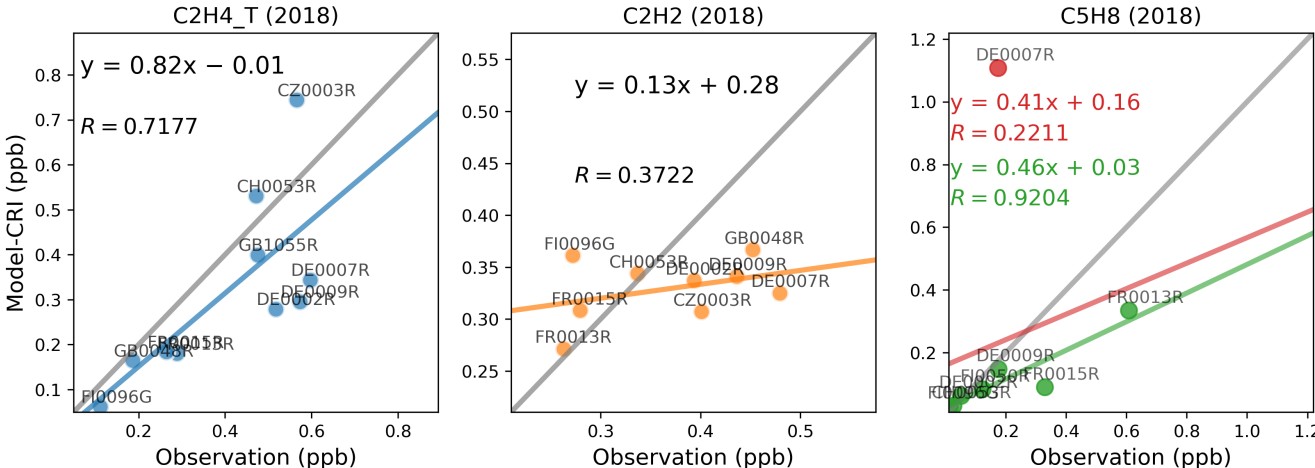

**Figure 12.** Scatter plots of annual mean modelled and measured ethene, ethyne, and isoprene concentrations in 2018. The term 'CRI' indicates that the model data is calculated using the CRIv2R5Em mechanism. In each plot, the grey line is the 1:1 line, and the other coloured line is the least-squares regression line. For isoprene, the outlier site is plotted in red; the red line is the regression line with the ourlier; the green line is the regression line without the ourlier.





**Table 8.** Summary of the comparison statistics between the model ($M$) and observation ($O$) for ethene and ethyne. $N$ is the number of sites. $R$ is the Pearson's correlation coefficient between annual means at various sites. Mean_O and Mean_M refer to the annual average concentrations (in ppb) of $O$ and $M$ over all sites, respectively. NMB is the Normalised Mean Bias, and NME is the Normalised Mean Error.

| C2H4_T | N | R | Mean_O | Mean_M | NMB | NME |
|---|---|---|---|---|---|---|
| Em-CEIP-2018 | 10 | 0.6472 | 0.405 | 0.287 | -29% | 40% |
| Em-CAMS-2018 | 10 | 0.7107 | 0.405 | 0.302 | -25% | 34% |
| CRI-CEIP-2018 | 10 | 0.6546 | 0.405 | 0.303 | -25% | 38% |
| CRI-CAMS-2018 | 10 | 0.7177 | 0.405 | 0.318 | -21% | 33% |
| CRI-CAMS-2019 | 7 | 0.7519 | 0.375 | 0.358 | -5% | 24% |
| **C2H2** | **N** | **R** | **Mean_O** | **Mean_M** | **NMB** | **NME** |
| Em-CEIP-2018 | 9 | 0.3628 | 0.368 | 0.321 | -13% | 20% |
| Em-CAMS-2018 | 9 | 0.3557 | 0.368 | 0.324 | -12% | 19% |
| CRI-CEIP-2018 | 9 | 0.3829 | 0.368 | 0.326 | -11% | 19% |
| CRI-CAMS-2018 | 9 | 0.3722 | 0.368 | 0.329 | -11% | 19% |
| CRI-CAMS-2019 | 9 | 0.5866 | 0.376 | 0.348 | -7% | 21% |



**Figure 13.** Time series comparisons of ethyne in 2018





### 4.2.2 Aromatic species

Figure 14 presents the comparisons of benzene, toluene, and o-xylene concentrations. Both the model and the measurements indicate that the concentrations of benzene and toluene are an order of magnitude higher than those of o-xylene. All three aromatic species demonstrate good model-measurement agreements.

For benzene, the NMB values are relatively small across the different model runs, whilst for toluene there is a moderate model underestimation of -33% to -39%, as presented in Table 9. The model's performance for o-xylene varies slightly between 2018 and 2019 comparisons, with better model-measurement agreement in 2018 than in 2019. However, only 4 valid sites are available in 2019, so the measurement data set is less representative. Furthermore, the observed o-xylene concentrations are so low (typically below 0.02 ppb) that these values may be artificially scattered due to uncertainties in the measurement.

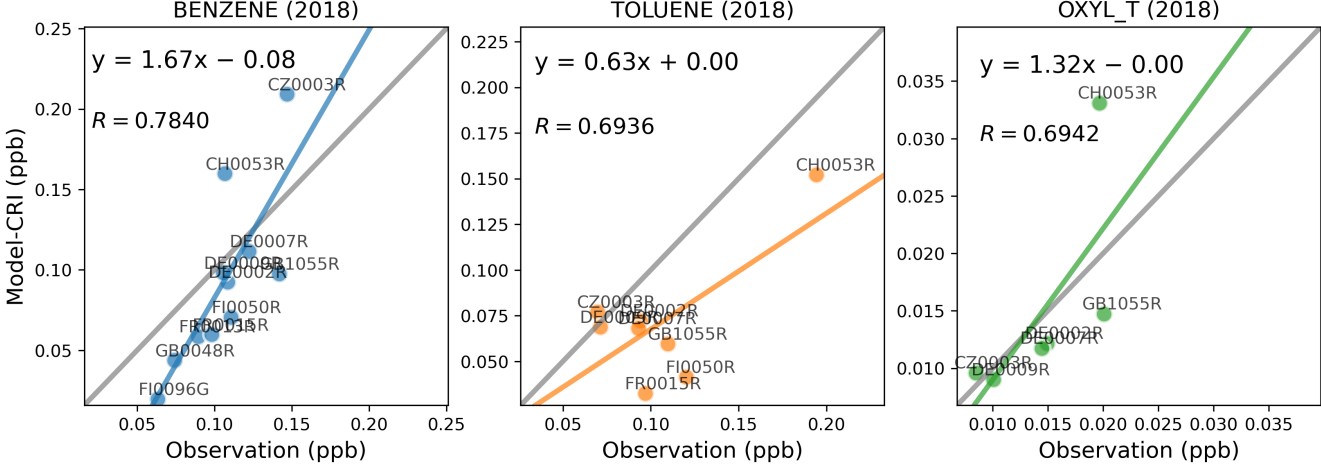

**Figure 14.** Scatter plots of annual mean modelled and measured benzene, toluene, and o-xylene concentrations in 2018. The term 'CRI' indicates that the model data is calculated using the CRIv2R5Em mechanism. In each plot, the grey line is the 1:1 line, and the other coloured line is the least-squares regression line.





**Table 9.** Summary of the comparison statistics between the model ($M$) and observation ($O$) for benzene, toluene, and o-xylene. $N$ is the number of sites. $R$ is the Pearson's correlation coefficient between annual means at various sites. Mean_O and Mean_M refer to the annual average concentrations (in ppb) of $O$ and $M$ over all sites, respectively. NMB is the Normalised Mean Bias, and NME is the Normalised Mean Error.

| BENZENE | N | R | Mean_O | Mean_M | NMB | NME |
|---|---|---|---|---|---|---|
| Em-CEIP-2018 | 11 | 0.7680 | 0.106 | 0.088 | -17% | 37% |
| Em-CAMS-2018 | 11 | 0.7858 | 0.106 | 0.092 | -14% | 33% |
| CRI-CEIP-2018 | 11 | 0.7666 | 0.106 | 0.090 | -16% | 36% |
| CRI-CAMS-2018 | 11 | 0.7840 | 0.106 | 0.093 | -13% | 32% |
| CRI-CAMS-2019 | 9 | 0.6761 | 0.102 | 0.089 | -12% | 33% |
| **TOLUENE** | **N** | **R** | **Mean_O** | **Mean_M** | **NMB** | **NME** |
| Em-CEIP-2018 | 8 | 0.6249 | 0.106 | 0.066 | -38% | 40% |
| Em-CAMS-2018 | 8 | 0.6938 | 0.106 | 0.069 | -35% | 36% |
| CRI-CEIP-2018 | 8 | 0.6290 | 0.106 | 0.068 | -36% | 38% |
| CRI-CAMS-2018 | 8 | 0.6936 | 0.106 | 0.071 | -33% | 34% |
| CRI-CAMS-2019 | 8 | 0.6481 | 0.115 | 0.070 | -39% | 39% |
| **OXYL_T** | **N** | **R** | **Mean_O** | **Mean_M** | **NMB** | **NME** |
| Em-CEIP-2018 | 6 | 0.7496 | 0.015 | 0.013 | -11% | 26% |
| Em-CAMS-2018 | 6 | 0.7038 | 0.015 | 0.014 | -3% | 31% |
| CRI-CEIP-2018 | 6 | 0.7365 | 0.015 | 0.014 | -6% | 25% |
| CRI-CAMS-2018 | 6 | 0.6942 | 0.015 | 0.015 | 3% | 30% |
| CRI-CAMS-2019 | 4 | 0.3953 | 0.019 | 0.017 | -13% | 43% |





### 4.2.3 VOC ratios: ethene/ethyne

Ethyne is widely acknowledged as an important tracer for combustion-related activities, particularly those connected to vehicular and residential sources. Ambient levels of ethyne typically rise during the winter, likely due to increased vehicle emissions and domestic heating (Dollard et al., 2007; Russo et al., 2010; McDonald et al., 2013). Consequently, numerous studies utilise VOC-to-ethyne ratios to identify combustion-related sources and to evaluate and constrain emission inventories (von Schneidemesser et al., 2023; Dominutti et al., 2020; Salameh et al., 2017).

Our findings show satisfactory model-measurement spatial correlations for ethene, but not for ethyne. A closer examination of the time series for these species shows that the modelled ethene concentrations (Fig. E3) align well with the observed temporal patterns, whereas the modelled ethyne concentrations (Fig. 13) are significantly lower during the winter months.

The modelled ratios of ethene to ethyne are influenced by these discrepancies between model and measurement. Table 10 presents both the measured and modelled ratios, calculated as the slope term of the least squares regression line between ethene 520 and ethyne, along with their corresponding linear regression coefficients for summer and winter, at available EMEP sites.

At most EMEP sites, linear correlations between measured ethene and ethyne are greater and ethene-to-ethyne ratios are higher (typically above 1) in winter compared to those in summer (typically below 1). The exception is the UK site at Auchencorth Moss, where the summer observations exhibit a larger ratio. The study by Boynard et al. (2014) reported an ethene-to-ethyne ratio of 2.78 in summer and 2.30 in winter in Paris during 2009-2010. In contrast, in Strasbourg there was a lower ratio 525 in summer (1.65) and a higher one in winter (2.01). These findings underscore that the dominant source of these two VOCs varies both seasonally and geographically.

In contrast to the measured data, the modelled ethene-to-ethyne ratios demonstrate weaker linear correlations and less pronounced seasonal patterns. The closest agreement between model and measurement is observed at the Beromünster site, where the model shows fairly good linear correlation coefficients of 0.71 in summer and 0.67 in winter. However, the modelled winter 530 ratio at this site (3.94) is more than twice the measured value (1.64). This discrepancy aligns with the previously noted underestimation of ethyne concentrations by the model during winter, as depicted in Fig. 13. At the rest of the sites, the modelled ratios diverge substantially from the measured values in both summer and winter. A detailed discussion on measurement issues for ethyne is presented in Sect. 4.4.

### 4.2.4 VOC ratios: benzene/ethyne

Similar discrepancies are also found for benzene to ethyne ratios. Table 11 shows the measured and modelled benzene-to-ethyne ratios and their corresponding linear regression coefficients in summer and winter.

The measured benzene concentrations are positively correlated with ethyne during summer with linear correlation coefficients typically above 0.40. In winter, the correlation is stronger with R value exceeding 0.89 at all sites, indicating common sources of the two species. The measured benzene-to-ethyne ratios at EMEP sites do not show large spatial nor seasonal vari-540 ations, varying around 0.2. Boynard et al. (2014) reported similar values in summer and winter with their benzene to ethyne ratios being 0.20-0.26 in Paris and 0.17-0.23 in Strasbourg.



**Table 10.** Ethene to ethyne ratios in 2018 at each site, calculated as the slope term of the least squares regression line between ethene(y) and ethyne(x). The linear correlation coefficient R of the regression is indicated in brackets.

| C2H4_T/C2H2 | Summer | | Winter | |
|---|---|---|---|---|
| | **Obs** | **Mod** | **Obs** | **Mod** |
| Beromünster | 1.04 (0.71) | 1.65 (0.71) | 1.64 (0.92) | 3.94 (0.67) |
| Kosetice | 0.60 (0.41) | 1.43 (0.71) | 1.67 (0.93) | 0.99 (0.08) |
| Waldhof | 0.23 (0.21) | 0.25 (0.44) | 1.76 (0.89) | -0.84 (-0.13) |
| Neuglobsow | 0.44 (0.45) | 0.15 (0.21) | 1.89 (0.97) | -0.27 (-0.02) |
| Zingst | 0.64 (0.80) | 0.03 (0.08) | 1.71 (0.95) | -3.11 (-0.40) |
| Pallas | 0.10 (0.12) | -0.02 (-0.14) | 1.07 (0.84) | -0.19 (-0.08) |
| Peyrusse Vieille | 0.07 (0.03) | -0.03 (-0.27) | 1.56 (0.85) | -1.98 (-0.27) |
| La Tardiere | 0.79 (0.43) | -0.05 (-0.37) | 1.08 (0.83) | 0.62 (0.14) |
| Auchencorth Moss | 1.11 (0.76) | 1.12 (0.63) | 0.66 (0.66) | 0.20 (0.06) |

By comparison, the modelled ratios vary seasonally at the Beromünster site. During summer, the modelled ratio of 0.33 approximates the measured value of 0.26 at this site. However, in winter, the modelled ratio of 0.98 is significantly higher than the measured winter ratio of 0.34. Given that Fig. E4 shows that the modelled benzene concentrations agree well with the mea-

545 sured values in both summer and winter, this is mainly caused by the model underestimation of winter ethyne concentrations at this site. Meanwhile, the negative linear correlation coefficients between modelled benzene and ethyne at most other sites point out significant deficiencies in the representation of ethyne emissions within current inventories. As shown in Table 11, in the majority of cases the correlations between benzene and ethyne are so poor that the value of their ratio becomes not relevant anymore. This is particularly concerning given that ethyne and benzene have comparable atmospheric lifetimes of 6-10 days

(Table. C1) , and are emitted from similar human activities, such as fuel consumption and combustion processes (Waked et al., 2016; Badol et al., 2008). In theory, this would result in similar spatial and temporal variation patterns for both species. If the model demonstrates a good spatial and temporal agreement with the measurement for benzene but fails to do so for ethyne — as observed in this study — it suggests that the problem may be specifically with the accuracy of ethyne emissions data. Such a discrepancy implies that the fundamental modelling of chemical reactions and transport processes is sound, but the emission

inputs need to be scrutinized and potentially revised to better reflect real-world conditions.

Furthermore, the model more accurately captures the linear correlation between ethene and ethyne, as well as between benzene and ethyne, only at the Beromünster site; at other locations the model's performance for these species is markedly poor. In such cases, drawing a comparison between modelled and observed ratios becomes impractical for most sites. The relative success at the Beromünster site for both ethene-to-ethyne and benzene-to-ethyne ratios, could provide insights for

potential enhancements in both modelling and measurement methodologies.

In summary, there appears to be a systematic underestimation of either the total emissions from the road transport and combustion-related sectors in the existing inventory or of the proportions of ethyne within these emissions. The current differ-





ences between the modelled and measured data indicate room for significant improvement, especially regarding ethyne sources during the winter months. More measurement evidence is therefore required to improve both the quantification of sector total emissions and the speciation within each sector.

**Table 11.** Benzene to ethyne ratios in 2018 which are calculated as the slope term of the least squares regression line between ethene(y) and ethyne(x). The linear correlation coefficient R is indicated in brackets.

| BENZENE/C2H2 | Summer | | Winter | |
|---|---|---|---|---|
| | **Obs** | **Mod** | **Obs** | **Mod** |
| Beromünster | 0.26 (0.82) | 0.33 (0.57) | 0.34 (0.96) | 0.98 (0.63) |
| Kosetice | 0.26 (0.47) | 0.35 (0.62) | 0.29 (0.93) | 0.46 (0.15) |
| Waldhof | 0.24 (0.77) | -0.00 (-0.01) | 0.30 (0.99) | -0.20 (-0.11) |
| Neuglobsow | 0.31 (0.88) | 0.00 (0.01) | 0.31 (0.99) | -0.02 (-0.01) |
| Zingst | 0.17 (0.83) | -0.03 (-0.13) | 0.30 (0.97) | -0.77 (-0.40) |
| Pallas | 0.11 (0.43) | -0.03 (-0.48) | 0.28 (0.94) | -0.04 (-0.06) |
| Peyrusse Vieille | 0.04 (0.09) | -0.07 (-0.58) | 0.24 (0.91) | -0.40 (-0.22) |
| La Tardiere | 0.25 (0.41) | -0.10 (-0.63) | 0.29 (0.93) | 0.17 (0.15) |
| Auchencorth Moss | 0.18 (0.76) | 0.08 (0.34) | 0.19 (0.89) | 0.03 (0.04) |

## 4.3 OVOCs

The assessment of the model's performance in predicting OVOC concentrations is constrained by the scarcity of available measurements. The EMEP regular monitoring network only have a few stations (often one or two) for a certain VOC and these stations do not measure the same set of OVOCs, so there is essentially no spatial information. Consequently, the EMEP IMP campaign for VOCs, conducted from 12 to 19 July 2022, serves as a valuable supplementary dataset. The following sections present comparisons with both regular 2018 measurements and the 2022 IMP campaign, utilising the CRIv2R5Em mechanism and the CAMS inventory.

### 4.3.1 Methanal and methylglyoxal in 2018

Figures 15 and F1 (in the supplement) show the time series of available measured and modelled methanal (also known as formaldehyde) and methylglyoxal, respectively. The sampling of both species is taken over a 4-hour period, as evidenced by the start and end times specified in the raw data. To facilitate a fair comparison with the measured data, the hourly model output at a specific station is averaged over the corresponding sampling duration.

The model successfully captures the temporal fluctuations of methanal and methylglyoxal at the FR0015R station through the year, and at the FR0013R station in the winter months. Generally, it tends to underestimate the peak concentrations during summer, particularly at the FR0013R station. However, the overall seasonal pattern and the concentration range of the model





data aligns reasonably well with the measurements, suggesting that the chemistry (and precursor emissions) associated with methanal and methylglyoxal are reasonably well-represented in the model.

**Figure 15.** Time series of methanal concentrations in ppb at two available monitoring sites in 2018. 'res:' denotes time-resolution; e.g. 'res: 3d' means that the measurements are conducted every three days (1w: one week). The letter 'H' denotes the site altitude.




### 4.3.2 Methanal and methylglyoxal in 2022 IMP

Whilst the 2022 IMP campaign was designed to produce high-resolution measurements, the OVOC measurement was still
carried out on a limited temporal scale. The majority of OVOCs are measured only once per day, and at inconsistent times
across different stations. Consequently, there are at most 10 data points available for a particular OVOC at a specific station,
although in many instances, the availability is further reduced to 3-5 data points due to the presence of invalid measurements. To
accommodate this limitation, average concentrations over the campaign period have been utilised to conduct linear correlation
analyses between contemporaneous model and measurement data.

Figure 16 shows the linear correlation relationships between modelled and measured concentrations of methanal and methyl-
glyoxal. The measured and modelled methanal concentrations align well, yielding a correlation coefficient of 0.91, despite a
moderate model underestimation. For such short time periods, this is excellent model-measurement agreement. As methanal is
a crucial intermediate in the oxidation of numerous other VOCs, this further illustrates that the model is effectively capturing
the overall photo-oxidation chemistry, despite the differing lumping processes applied to various VOC groups. Moreover, the
model underestimation for HCHO during the IMP, which took place in July 2022, is consistent with the model underestimation
at some sites in July 2018 shown in Fig. 15. Inaccuracy in modelled HCHO photolysis rate could be one explanation, which
would manifest more in summer.

Similar model underestimation is also observed for methylglyoxal (Fig. 16 ). Atmospheric sources of methylglyoxal are
multiple and include direct emissions from, for example, industrial emissions, vehicle exhausts, and biomass burning, and
secondary formation from the oxidation of biogenic and anthropogenic precursors (e.g., isoprene, aromatics) (Stavrakou et al.,
2009; Rodigast et al., 2016; Li et al., 2022). It is possible that the model overestimates the rate of photolytic loss for this period
in July, when solar flux is at its maximum. van Caspel et al. (2023) report that the EMEP model's $J$-values for methylglyoxal
are larger than the observed value by a factor of 2 at the Chilbolton site during wintertime, highlighting the large degree of
uncertainty in this photolysis rate. Given that photolysis is an important loss mechanism for methylglyoxal (Chen et al., 2000),
and given that it is likely that the overestimated modeled $J$-values also occur during summertime, this has the potential to cause
the EMEP model to underestimate methylglyoxal concentrations during the IMP.

### 4.4 Discussions on measurement issues

The evaluation of model performance is inherently constrained by the uncertainties in the emissions, the model, and ose as-
sociated with the measurements. It's essential to recognise that any analysis comparing modelled and measured data will
have limitations due to these uncertainties. A robust evaluation necessitates high-quality measurements. Owing to the exten-
sive chemical variation across VOC species, no singular analytical method is capable of identifying all atmospheric VOCs.
Consequently, various methods must be employed, introducing variability in speciation, temporal resolution, and analytical
uncertainties across datasets. For example, many VOC inter-comparison studies have revealed significant discrepancies among
participating laboratories, especially at low concentrations and when analysing real-world air samples as opposed to synthetic
calibration gas mixtures ((Ballesta et al., 2001; Slemr et al., 2002; Apel et al., 2003; Plass-Dülmer et al., 2006)).





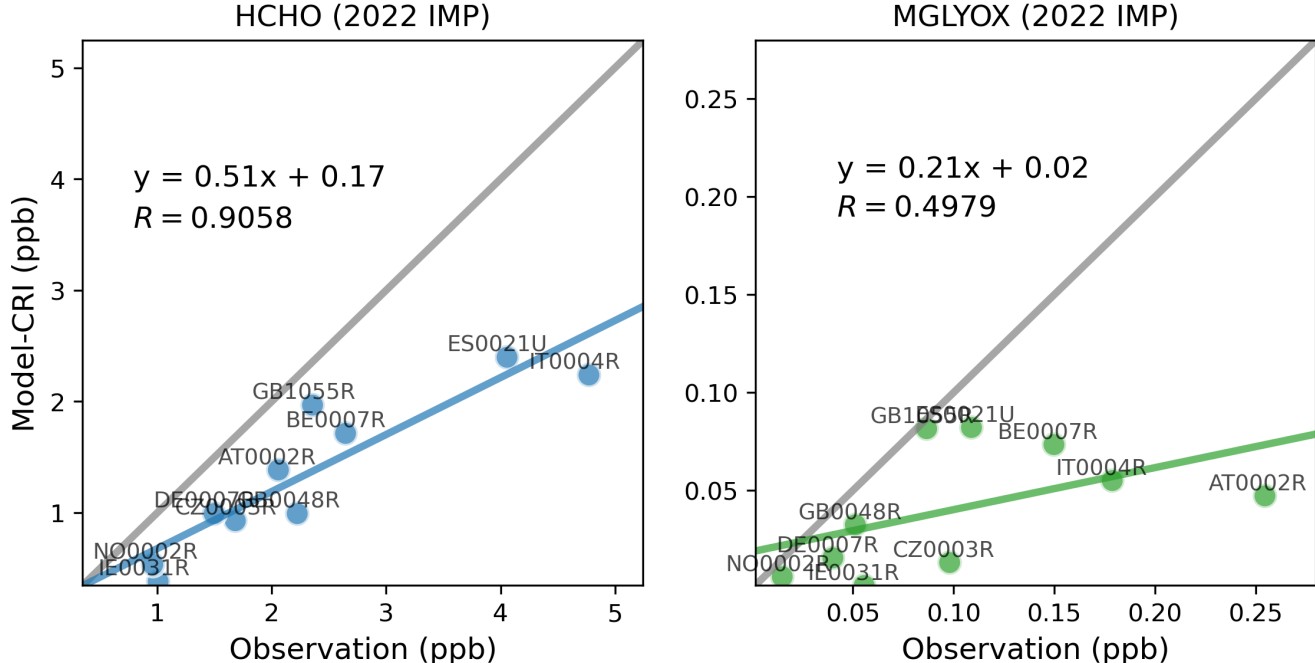

**Figure 16.** Scatter plots of average modelled and measured methanal and methylglyoxal concentrations during 2022 IMP.

In particular, the accurate determination of ethyne presents a distinct challenge relative to other atmospherically significant VOCs such as propane, butane, isoprene, and benzene (Badol et al., 2004; Rappenglück et al., 2006; Hoerger et al., 2015). The observed poor spatial correlation between model and measurement for ethyne in this study is speculated to be at least partly attributable to inconsistencies in methodology across different laboratories. Hoerger et al. (2015) pointed out that variations in laboratory performance could arise from two key factors. Firstly, a loss of ethyne due to breakthrough in the adsorption trap; secondly, different analytical systems may respond inconsistently to the same calibration standards. These factors contribute to significant discrepancies between the measured and pre-assigned concentrations of ethyne in mixed hydrocarbon standards. Moreover, insufficient characterisation of the analytical system's response to both dry calibration standards and humid real-world air samples can introduce significant biases in the reported concentrations.

Furthermore, Hoerger et al. (2015) and Plass-Dülmer et al. (2006) reported artefacts affecting the measurement of alkenes, albeit arising from different sources. Hoerger et al. (2015) identified that instruments employing a Nafion® Dryer to remove humidity produced blank values up to 0.35 ppb for $C_2$–$C_3$ alkenes. These blank values necessitate subtraction during either calibration or ambient air measurements to ensure accuracy. On the other hand, Plass-Dülmer et al. (2006) observed that the use of canisters led to increased concentrations of alkenes. This phenomenon was attributed to slow production of alkenes from the inner walls of the canisters themselves. Collectively, these artefacts could lead to systematic biases in measured concentrations



of alkene species, and explain, at least in part, the discrepancies between measured and modelled concentrations observed in this study.

Last but not least, although numerous inter-comparisons of NMHC measurements have been conducted over the years, assessments of measurement accuracy and consistency for OVOCs such as alcohols, aldehydes, and ketones, are notably sparse. Recently, a few OVOCs intercomparison projects have commenced within ACTRIS. However, as of now, no official reports have been released. At present, our model evaluation of OVOCs is very limited in scope, being able to focus on two OVOC species measured at only a few stations or over brief periods. Statistically speaking, the available measurement datasets are insufficient for drawing robust conclusions. Compounds like methanal and methyglyoxal have both primary anthropogenic and biogenic sources and are also commonly generated as oxidation products from other VOCs. This complexity makes it challenging to determine whether the model's underestimation of these compounds arises from missing primary emissions, underrepresented secondary chemical production, overestimated chemical and deposition loss, or possibly a combination of all these. In light of the general increase in domestic solvent consumption and the growing use of alcohols in fuels (UK-AQEG, 2020; Whalley et al., 2018; Dunmore et al., 2016), the relative abundance and subsequent impact of OVOCs compared to NMHCs on ozone and aerosol chemistry are likely to become increasingly significant in future. Therefore, there is a pressing need for more robust, long-term, and multi-station OVOC monitoring efforts moving forward.

In summary, it is essential to more accurately characterise and quantify the uncertainty associated with individual VOC measurements. It is imperative to harmonise analytical procedures, particularly in relation to real-world air sampling methods. Concurrently, the implementation of more stringent quality assurance and quality control checks is crucial, akin to the procedures being developed within ACTRIS. This would not only ensure the submission of high-quality measurement data to public data repositories and end-users but also facilitate the development of more precise VOC emission speciations. Such advancements would, in turn, contribute to achieving a higher degree of agreement between modelled and measured data.

### 4.5 Impacts of changing inventories and mechanisms on model performance

One of the biggest challenges in accurately representing VOCs in atmospheric chemistry models lies in the manner in which these compounds are reported by emission inventories. Typically, VOCs emissions are presented as aggregate values, necessitating the use of sector-specific speciation profiles to apportion these lumped masses into individual VOC species. Achieving good agreement between the model and the measurement therefore depends on the accurate estimation of two factors: total VOC emissions and VOC speciation. However, it remains an open question as to whether one of these factors holds greater importance than the other in determining the accuracy of the modelled VOC concentrations.

Our model experiments offer an opportunity to address this question. The two emission inventories utilised in this study report slightly differing total emissions for individual sectors. For example, the largest emitting sector in the CEIP inventory is sector-F Road Transport (30% of its total, similarly hereinafter), followed by sector-D Fugitive (22%) and sector-E Solvents (14%) emissions (Fig. 2). By comparison, the CAMS inventory identifies solvents (26%) as the sector with the largest total emissions, with Road Transport (22%)—comprising four sub-sectors, each with distinct profiles—coming second. The speciation profile of sector-F Road Transport used for the CEIP inventory is derived from the individual speciation profile of F1, F2,




F3, F4 and sector totals of these sub-sectors reported by the CAMS inventory (Sect.3.3.4). The emission profiles for all other sectors are identical between the two inventories.

Results from Sect. 4 reveal that model simulations based on the two emission inventories yield very similar statistical metrics. Table 12 summarises the descriptive statistics of linear correlation coefficients between modelled and measured annual average concentrations of the 12 NMHCs investigated in this study (i.e., 12 species listed in Table 6-9). In general, model simulations

using the CAMS inventory show slightly better agreements with measurements than those using the CEIP inventory. Using the CRIv2R5Em mechanism in 2018, the mean correlation coefficient is 0.59 for CEIP and 0.64 for CAMS. Moreover, both inventories result in model overestimation of n-butane and n-pentane but underestimation of i-butane and i-pentane, and is due to the large emissions assigned to n-butane and n-pentane isomers from the solvent sector. This evidence suggests that the emission profiles are likely to exert a more significant impact on the agreement between modelled and measured data than the

total emissions reported for each sector. Therefore, the future focus may need to shift towards refining these speciation profiles to improve model accuracy.

Compared to 2018, the model's performance in the 2019 simulation improves for some VOCs but deteriorates for others. For instance, the correlation coefficient for ethane in 2019 is 0.70, compared to 0.58 in 2018. Conversely, for o-xylene, the correlation coefficient decreases to 0.39 in 2019 from 0.69 in 2018. As previously discussed, such variations in model perfor-

680 mance between the two years can be attributed to changes in the specific stations and the amount of valid sites available in each year, as well as in meteorology and local emissions. More importantly, for most species, the model performance does not significantly differ between the two years, as evidenced by the similar statistical data presented in Table 12 and previous tables.

**Table 12.** Summary of linear correlation coefficients between modelled and measured annual average concentrations of the 13 discussed NMHCs in different model runs.

| $R$ | Min | Max | Median | Mean |
|---|---|---|---|---|
| Em-CEIP-2018 | 0.3343 | 0.9100 | 0.6126 | 0.5930 |
| Em-CAMS-2018 | 0.3557 | 0.9331 | 0.6534 | 0.6382 |
| CRI-CEIP-2018 | 0.3263 | 0.9105 | 0.6126 | 0.5932 |
| CRI-CAMS-2018 | 0.3722 | 0.9329 | 0.6514 | 0.6375 |
| CRI-CAMS-2019 | 0.3953 | 0.8866 | 0.6603 | 0.6517 |

Finally, the discrepancies between model outputs using the two different chemical mechanisms, EmChem19rc and CRIv2R5Em, are negligible. Utilising the CAMS inventory in 2018, the mean linear correlation coefficient across the 12 VOCs is 0.64 for

both mechanisms. This further demonstrates that the performance of EmChem19rc is comparable to that of CRIv2R5Em, and that the choice of chemical mechanism amongst those used here does not substantially affect the model-measurement agreement. Such close alignment demonstrates the robustness of the overall chemistry and transport processes parameterised within the EMEP model.



## 4.6 Impacts of changing emission speciation on modelled ozone concentrations

A relevant question is the extent to which regional transport model results are sensitive to details of the VOC speciation. To a first approximation, the ozone production from a VOC is proportional to the amount of VOC that has time to react with OH or other oxidants. Thus, close to source areas, the fast-reacting VOC such as ethene make immediate contributions to ozone, but these VOC are quickly consumed. Further downwind, other species such as alkanes are degraded and their contribution builds up. Thus, metrics of ozone production such as POCP (which uses ethene as the reference value of 100%) show that

over short time periods alkanes have low POCPs, but over longer time-periods and greater geographical extent the POCP of alkanes become significant (Andersson-Sköld et al., 1992; Stockwell et al., 2001). For example, Simpson (1995) calculated POCP values of n-butane from ca. 30% in NW Europe to ca. 80% in southern and eastern areas. Although a full evaluation of the impacts of uncertainties in VOC speciation on ozone is beyond the scope of this study, we have compared two model runs for 2018:

1. nDef - default VOC speciation for this study, using the EmChem19rc chemistry mechanism and the CAMS inventory as described in Sect. 3.3.

    2. Sol6 - in which the same chemistry mechanism and emission inventory are used, but the VOC speciation of solvent sector has been replaced by that of gasoline vehicles (exhaust, sector F1, emep code 6 - see Table B1).

The second run, Sol6, is purely for illustration, but lets us examine the extent to which the speciation of this very uncertain

sector (c.f. Oliveira et al., 2023) matters. The use of the F1 speciation provides a more reactive mixture than our default solvent splits. Both runs have been conducted using the CAMS-REG emissions, $0.1° \times 0.1°$ resolution, and for 2018. Fig. 17 shows the modelled $O_3$ from these two model runs, along with the difference. It can be seen that on most days the change of speciation makes little difference to the modelled $O_3$, but changes of up to 9 ppb are calculated. Comparing with 103 sites (altitude <1000 m) from the full EMEP network, the overall statistics are remarkably unchanged: normalised mean bias is 4% for nDef

and 5% for Sol6, spatial correlation coefficients are 0.85 and 0.84 respectively, and temporal correlation coefficients are 0.95 for both runs.

In the Supplementary, Sect.G, we illustrate changes in ozone and associated metrics at the European scale, and with a focus over Madrid in Spain. These comparisons confirm that changes in VOC speciation have little impact on mean ozone levels, but changes can be significant close to major NOx sources.





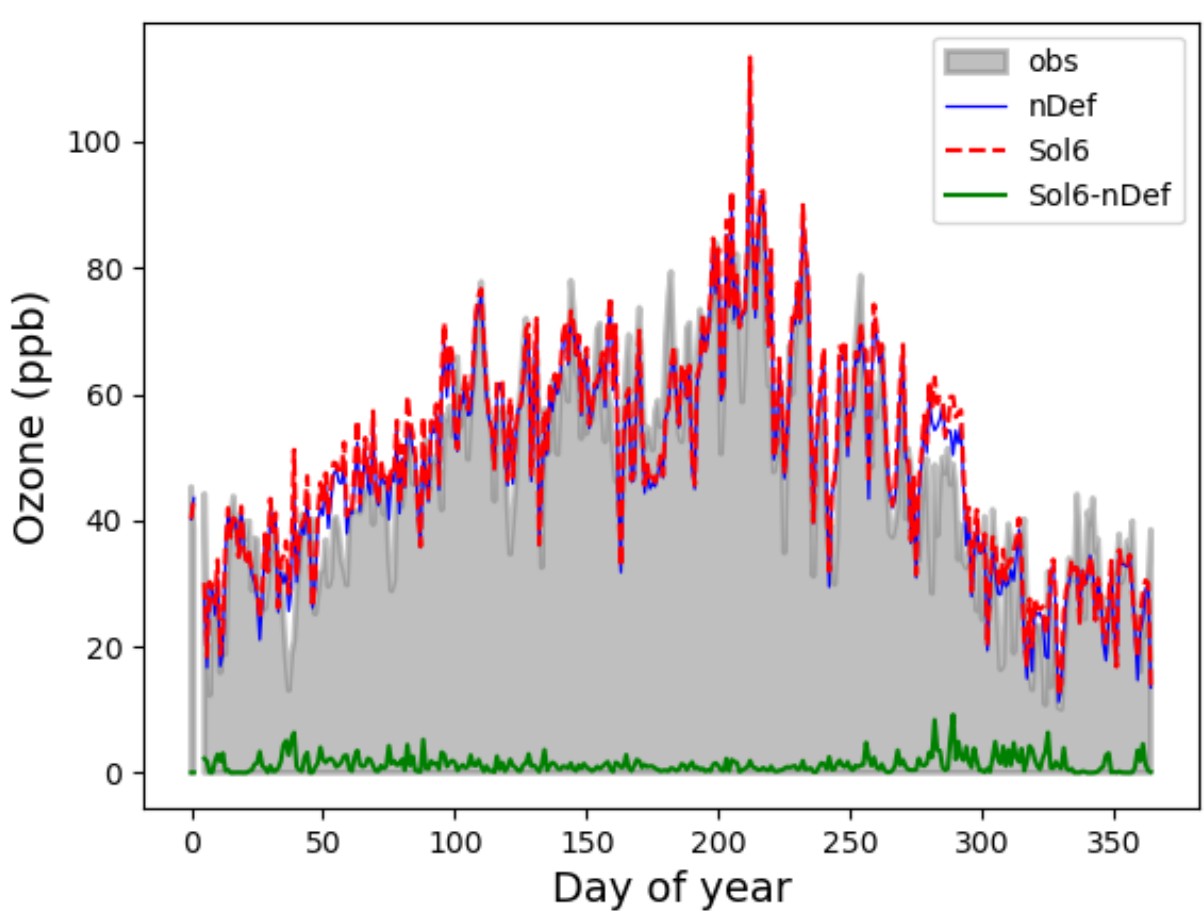

**Figure 17.** Impacts of VOC sensitivity tests on modelled daily maximum $O_3$ at Beromünster. nDef and Sol6 are two model runs, and the lowest line gives the difference, Sol6-nDef. Observed $O_3$ shown by shaded area. Model runs for 2018.





## 5 Conclusions

This model evaluation study is the first intensive comparison of VOCs between the EMEP model and measurements for many years. Considering that the composition of VOCs has undergone significant changes over the past several decades, and that there is lack of evaluation studies, we are keen to know how accurately these real-world changes in VOC profiles are captured in recent emission inventories, and how well the model's VOC concentrations agree with measured values.

To address these research questions, a comprehensive spatial and temporal evaluation of model outputs with VOC measurements from the EMEP network was carried out for the year 2018 and 2019, and for the IMP campaign in summer 2022. Both CEIP and CAMS emission inventories were utilised, along with two different chemistry mechanisms—EmChem19rc and CRIv2R5Em. To model pure VOC concentrations for comparison with measurement data, we have developed a detailed VOC emission speciation for all EMEP sectors based on data sourced from the UK NAEI, EEA emission inventory guidebook, and several academic studies.

The degree to which the modelled and measured VOCs agree varies depending on the specific species, suggesting potential issues with the boundary conditions and emission speciation for these species. In general, the model successfully captures the overall spatial and temporal variations of major alkanes such as ethane and n-butane, but less so for propane and iso-butane.

The model's underestimation of propane concentrations and the smaller propane-to-ethane ratios compared to measurements are likely caused by a combination of issues with the boundary conditions and potential missing propane emissions from the oil, natural gas and LPG sectors in current inventories.

Interestingly, the model overestimates n-butane and n-pentane while underestimating iso-butane and iso-pentane. Further analysis of the ratios among the butane and pentane isomers reveals that both model and measurement data exhibit strong linear correlations between iso-butane and n-butane, as well as between iso-pentane and n-pentane, with correlation coefficients typically exceeding 0.8. This suggests common sources for these pairs of butane and pentane isomers. However, modelled ratios of iso- to n-butane and iso- to n-pentane are approximately one-third of the measured ratios. Given that iso-butane and n-butane have similar atmospheric lifetimes (as do iso-pentane and n-pentane), such a discrepancy in their ratios likely stems from differences in their emissions. Indeed this disparity is largely driven by significant emissions of n-butane and n-pentane from the Solvent sector. It is possible that emissions of iso-butane and iso-pentane in the speciation profile of the Solvent sector may be underrepresented, or that the total emissions from transport activities and fuel evaporation are higher than what is currently included in the emission inventories. Alternatively, both scenarios might be true.

For unsaturated NMHCs, results are very mixed, with good results for ethene and aromatics, but very poor results for ethyne. In addition, the model underestimates ethyne concentrations significantly during winter. The different model performance strongly points to shortcomings in the spatial and temporal patterns and magnitudes of ethyne emissions. The modelled ethene-to-ethyne and benzene-to-ethyne ratios differ significantly from measured ratios. In general, most EMEP sites display stronger linear correlations and smaller VOC-to-ethyne ratios, while the model data shows poor correlations and therefore the modelled ratios become impractical for most sites.



For OVOCs, methanal and methylglyoxal demonstrate reasonably good agreement between modelled and measured time series throughout the year 2018 simulations, though both are underestimated in the 2022 IMP campaign. As both species have significant secondary sources from the oxidation of numerous other VOCs, this further illustrates that the model is effectively capturing the overall photo-oxidation chemistry processes. Additionally, it is also important to note that the lack of measurement data seriously limits the evaluation of other OVOC species.

Generally, simulations that employed the CAMS inventory displayed slightly better comparison results for certain VOCs compared to those utilizing the CEIP inventory, which is likely due to the inclusion of more detail in the road traffic emissions sectors (F1–F4) in CAMS. Moreover, despite the two inventories reporting somewhat different sector total emissions, the overall similar model performance demonstrates that the speciation profiles are likely to exert a more significant impact on the model-measurement agreement. Therefore, the future focus may need to shift towards refining these speciation profiles to improve the emission accuracy.

In summary, the model seems to do a reasonable job of capturing spatial patterns and time series of some VOC species (e.g. n-butane, longer-chain alkanes, aromatics, HCHO), but performs less well for others (e.g. propane, ethyne). Such discrepancy in model performance indicates potential issues pertaining to certain VOC emissions and to the model setup of boundary and initial conditions. It would be beneficial to engage in further discussions with measurement teams to possibly incorporate insights from measurement data to refine the emission speciation applied in the model. Despite certain limitations in model-measurement comparisons, the detailed evaluations in this study support the use of the EMEP model for analysing the significance of different types of VOCs to ozone and aerosol formation, and illustrate the benefits of the VOC measurement data for model and emissions evaluation. Moreover, this study also provides a valuable reference for VOC speciation and evaluation in other modelling studies.

*Code and data availability.* As described and referenced in Sect. 3.1 of this paper, this study used the latest EMEP MSC-W model version rv5 (Simpson et al., 2023), with source code available at MSC-W (2023) (last access: 20th December 2023). The VOC tracers and their related code are integrated into the EMEP model via the GenChem system (Simpson et al., 2020), with source code available at MSC-W (2020) (last access: 20th December 2023). All measurements are available on the platform of EBAS (2023) (last access:20th December 2023). The model outputs and measurement data presented in the figures and tables in this paper as well as the corresponding Python scripts are available at Ge (2023).

*Author contributions.* YG, DS, and SS conceptualized and designed the study. YG wrote the initial manuscript and undertook model simulations, downloaded and prepared measurement data sets, and visualised the results of model-measurement comparisons. DS contributed to the development of chemistry mechanisms and provided modelling support. SS contributed to the checking and the interpretation of VOC measurements. MRH contributed to the discussion materials and editing of the manuscript. WVC contributed to the Cloud-J related modelling and data interpretation. SR, BH, and TS contributed to the proofreading and discussion materials on measurements. All authors provided review comments and approval of the final version.



*Competing interests.* The authors declare that no competing interests are present.

*Acknowledgements.* We acknowledge all data providers and the great efforts of EMEP and ACTRIS to make long-term measurements public and available, and for the work with quality assurance and quality control of VOC data in Europe. All measurement data used in this study are obtained from the EBAS database.

The Norwegian Ministry of Climate is also greatly acknowledged for supporting both model and database development at the EMEP
Centres, MSC-W and CCC. Measurements in Beromünster were supported by FOEN (Swiss Federal Office for the Environment) and by ACTRIS-CH. We are also grateful to Adéla Holubová and the whole measurement team from Czech Hydrometeorological Institute, for their contribution to the measurement data in Košetice. We also extend our gratitude to the UK NAEI team for supplying the speciated VOC emissions data. Our thanks are due as well to Antoon Visschedijk, Jeroen Kuenen, and the TNO team for their valuable assistance with the VOC speciation of agricultural emissions.

We also acknowledge the use of Pyaerocom version v0.15.0 (AeroCom, 2023) for the comparison of EBAS data against simulations in Section 4.6.



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



## Appendix A: EmChem19rc species

**Table A1.** Summary of current primary VOC species in EmChem19rc. Species coloured in blue are newly added VOC tracers; green indicates tracers for existing lumped surrogates; orange indicates species that have secondary production from lumped surrogates. TBUT2ENE represents 2-butene; NPROPOL_T and IPROPOL_T represent 1-propanol and 2-propanol respectively; GLYOX and MGLYOX represent glyoxal and methylglyoxal respectively; MEK represents methyl ethyl ketone. XTERPENE is a lumped surrogate for other biogenic species.

| EmChem19rc | Species | | | |
|---|---|---|---|---|
| Shorter-chain alkane | C2H6_T | C3H8_T | NC4H10_T | IC4H10_T |
| Longer-chain alkane | NC5H12_T | IC5H12_T | NC6H14_T | NC7H16_T |
| Alkene | C2H4_T | C3H6_T | TBUT2ENE_T | |
| Alkyne | C2H2_T | | | |
| Aromatics | BENZENE | TOLUENE | OXYL_T | |
| Alcohol | CH3OH | C2H5OH_T | NPROPOL_T | IPROPOL_T |
| Aldehyde | HCHO | CH3CHO | | |
| Dialdehyde | GLYOX | MGLYOX | | |
| Ketone | CH3COCH3_T | MEK | | |
| Carboxylic acid | HCOOH_T | CH3CO2H_T | | |
| Biogenic VOC | C5H8 | $\alpha$-PINENE | $\beta$-PINENE_T | XTERPENE |
| Rest[†] | OTH_ALKANE_T | | | |

Notes †: Rest includes other alkanes and some other species.





## Appendix B: Emission sectors and time factors

Table B1 details the 19 emission sectors incorporated in the model, which comprise 13 GNFR sectors and additional 6 subsectors as defined by the CAMS emission inventory. CEIP provides emission data from these 13 GNFR sectors, treating sectors A and F collectively (i.e., without individual data for subsectors A1, A2, F1, F2, F3, F4). By contrast, CAMS reports emissions from the same 13 GNFR sectors, but it divides the emissions from sectors A and F into their respective subsectors. Figs. B1 and B2 illustrate the monthly factors for two countries, the UK and Switzerland, see Sect. 3.3.

**Table B1.** Relations between EMEP, GNFR_CAMS and SNAP sectors

| EMEP code | SNAP | GNFR_CAMS code | Source |
|---|---|---|---|
| 1 | 1 | A | Public Power |
| 2 | 3 | B | Industry |
| 3 | 2 | C | Other Stationary Combustion |
| 4 | 5 | D | Fugitive |
| 5 | 6 | E | Solvents |
| 6 | 7 | F | Road Transport |
| 7 | 8 | G | Shipping |
| 8 | 8 | H | Aviation |
| 9 | 8 | I | Offroad |
| 10 | 9 | J | Waste |
| 11 | 10 | K | Agri - Livestock |
| 12 | 10 | L | Agri - Other |
| 13 | 3 | M | Other |
| 14 | 1 | A1 | PublicPower & Point |
| 15 | 1 | A2 | PublicPower & Area |
| 16 | 7 | F1 | Road Transport Exhaust - Gasoline |
| 17 | 7 | F2 | Road Transport Exhaust - Diesel |
| 18 | 7 | F3 | Road Transport Exhaust - LPG |
| 19 | 7 | F4 | Road Transport Non-Exhaust |

Notes: The EMEP codes 1–19 are used in the EMEP model. The SNAP codes 1–11 are from the earlier EMEP model version.



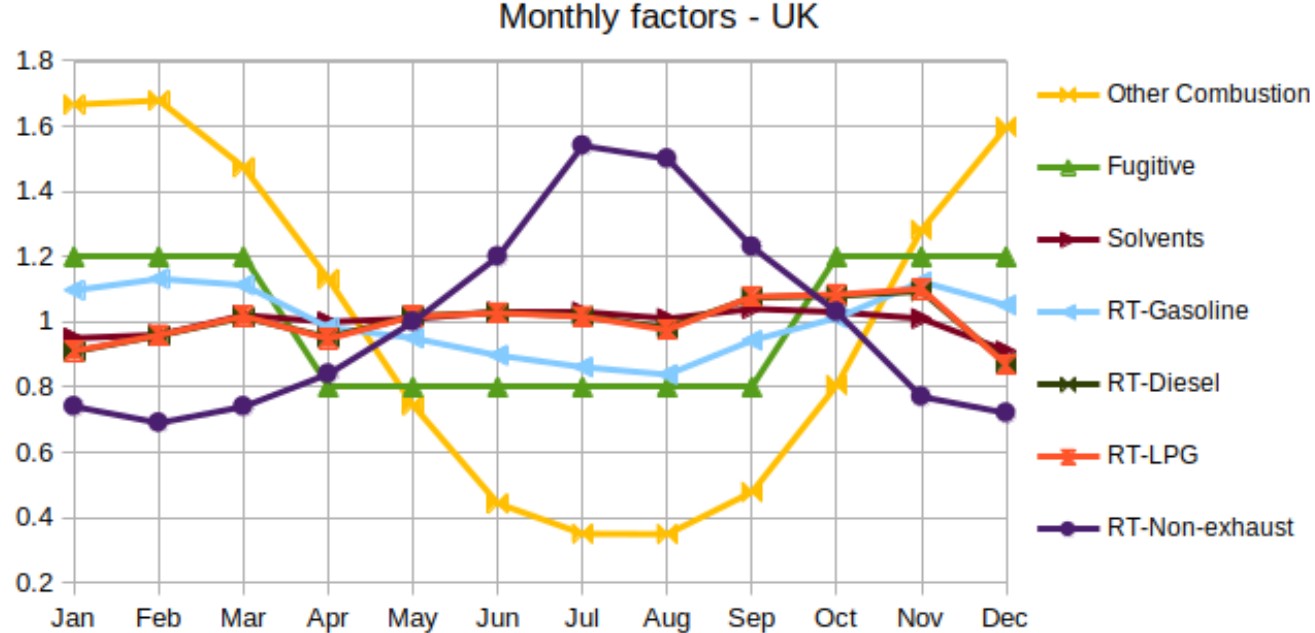

**Figure B1.** Monthly factors of VOC emissions for the UK, derived from CAMS_TEMPO (Guevara et al., 2021).

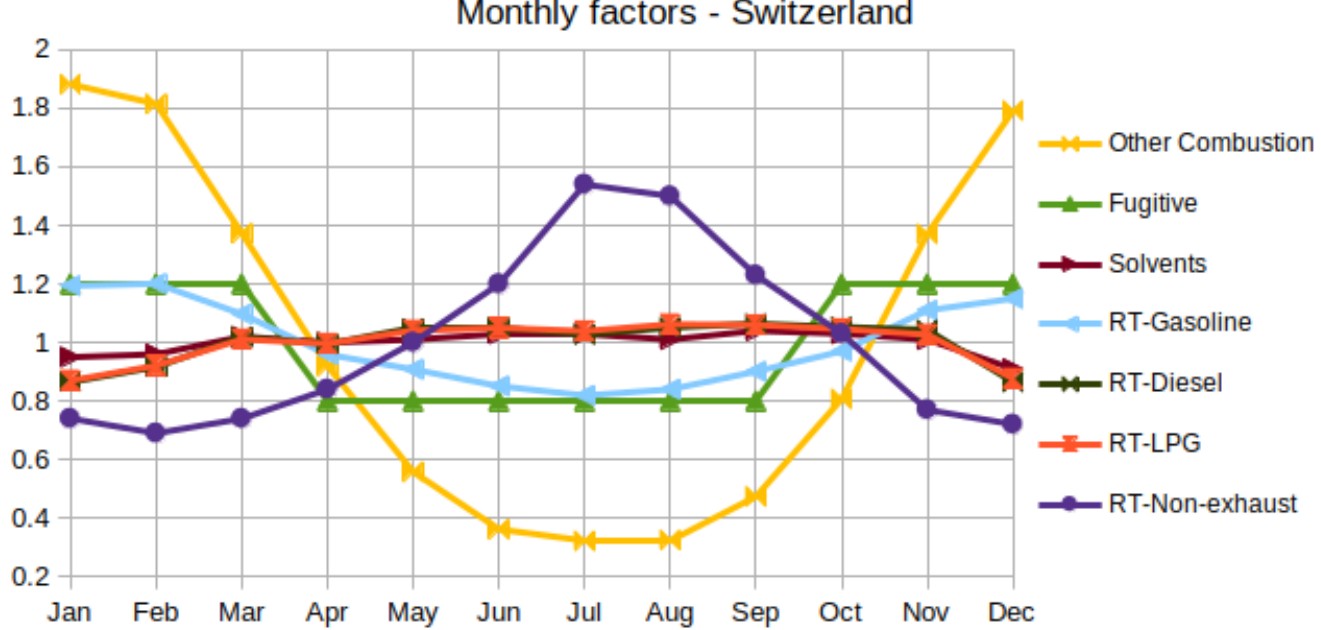

**Figure B2.** As Fig. B1, but for Switzerland



## Appendix C: Lifetimes of VOCs

**Table C1.** Atmospheric lifetimes of selected VOCs, calculated using CRIv2R5Em rates (the same as MCM rates), and obtained from Seinfeld and Pandis (1998), assuming OH of $1.5 \times 10^6$ molecules cm$^{-3}$ (12 hour daytime average), 30 ppb O$_3$ (24h average), 1 ppt NO$_3$ (12h night average).

|  | Lifetime due to: | | | |
|---|---|---|---|---|
|  | OH | O$_3$ | NO$_3$ | $h\nu$ |
| ethane[†] | 32 d | | | |
| propane[†] | 7.2 d | | | |
| n-butane[†] | 3.28 d | | n.e. | |
| i-butane[†] | 3.52 d | | | |
| n-petane[†] | 1.93 d | | | |
| i-petane[†] | 2.09 d | | | |
| n-hexane[†] | 1.42 d | | | |
| n-heptane[†] | 1.10 d | | | |
| ethyne[†] | 10.2 d | | | |
| ethene[†] | 5.8 min | 10.1 d | 6.1 yr | |
| isoprene[†] | 1.85 h | 1.23 d | 17.3 h | |
| benzene[†] | 6.3 d | | | |
| toluene[†] | 1.37 d | | n.e. | |
| o-xylene[†] | 13.6 h | | n.e. | |
| n-butane | 5.7 d | | 2.8 yr | |
| benzene | 12 d | | | |
| toluene | 2.4 d | | 1.9 yr | |
| m-xylene | 7.4h | | 200 d | |
| HCHO | 1.5 d | | 80 d | 4 h |
| C5H8 | 1.7 h | 1.3 d | 0.8 h | |

[†] calculated from CRIv2R5Em rates, for 25°C, with n.e. denoting not estimated (reactions are not included in the mechanism).



## Appendix D: Alkanes comparisons

**Figure D1.** Time series of modelled and measured ethane concentrations in 2018.



**Figure D2.** Time series of modelled and measured propane concentrations in 2018.



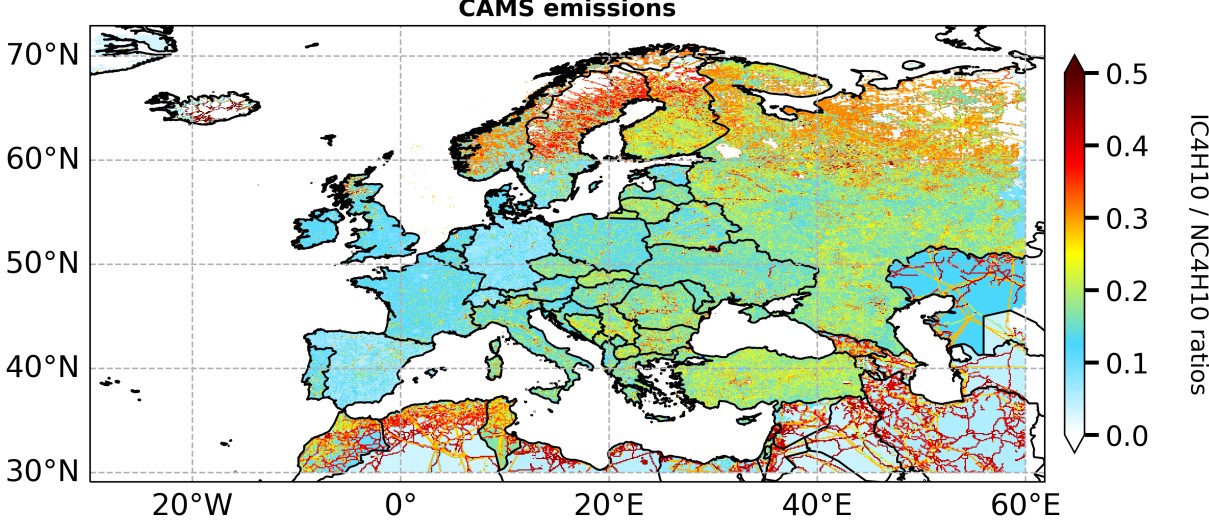

**Figure D3.** Map of iso-butane to n-butane emitting ratios using the CAMS emission inventory





1150 **Appendix E: Unsaturated NMHCs**

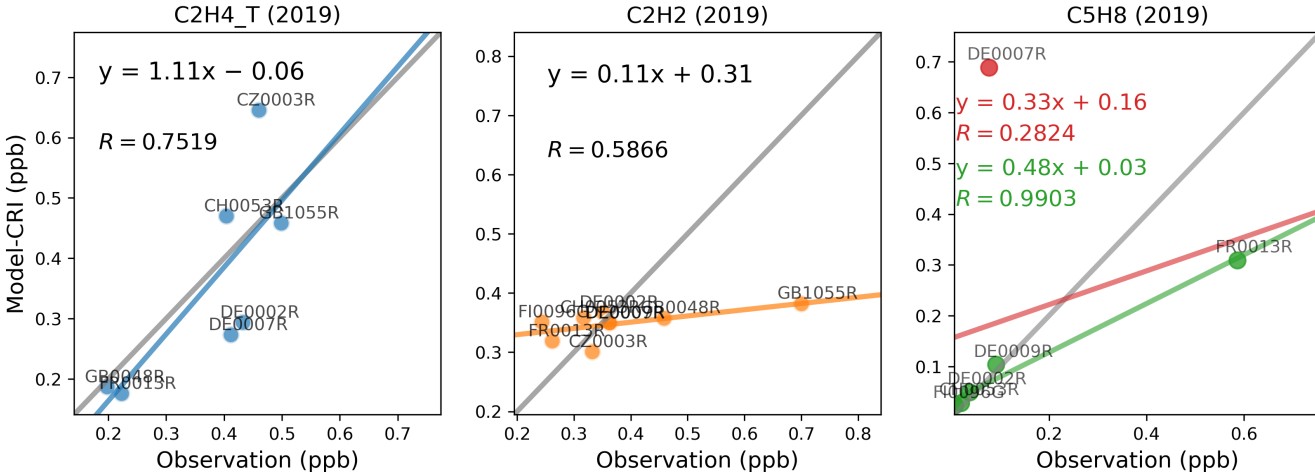

**Figure E1.** Scatter plots of annual mean modelled and measured ethene, ethyne, and isoprene concentrations in 2019. The term 'CRI' indicates that the model data is calculated using the CRIv2R5Em mechanism. In each plot, the grey line is the 1:1 line, and the other coloured line is the least-squares regression line. For isoprene, the outlier site is plotted in red; the red line is the regression line with the outlier; the green line is the regression line without the ourlier.





**Figure E2.** Time series of modelled and measured isoprene concentrations in 2018.





**Figure E3.** Time series of modelled and measured ethene concentrations in 2018.





**Figure E4.** Time series of modelled and measured benzene concentrations in 2018.





## Appendix F: OVOCs

**Figure F1.** Time series of modelled and measured methylglyoxal concentrations in 2018.



**Appendix G:  Impacts of changing emission speciation on modelled ozone concentrations**

As discussed in Sect. 4.6, we have made model calculations with both a reference run (nDef), and a sensitivity test (Sol6) in which the VOC speciation of solvents was replaced by the more reactive mix from GNFR sector F1. We have investigated changes in mean of daily maximum ozone (MDMax $O_3$), the 4th highest daily maximum 8-hour ozone (4MDA8), and in the highest daily maximum 8-hour ozone (1MDA8). Figure G1(a) and (b) shows that this change of speciation has only small impacts on MDMax $O_3$ and 4MDA8, with changes below 1 ppb in the majority of areas. The 1MDA8 values, Fig. G1(c), show a bigger response, of more than 5 ppb in some areas, but in very localised regions such as southern UK, the Po Valley in northern Italy, or near Madrid in Spain. These areas coincide with areas of high NOx emissions as expected - ozone chemistry is most sensitive to VOC emissions in such regions (Seinfeld and Pandis, 1998).

Figure G2 illustrates some of these changes for the Madrid region in more detail. The distribution of MDMax $O_3$, Fig. G2(a), shows maximum values north of the city centre, with much lower values to the east - these areas correspond to the high NOx emission areas (Fig.G2(e)). The changes in MDMax $O_3$, 1MDA and 4MDA are shown in Fig.G2(b)-(d). It is notable that VOC impacts can be significant (here over 20 ppb for 4MDA8 and 1MDA8), but are restricted to grid squares within 20–30 km of the major emission sources.



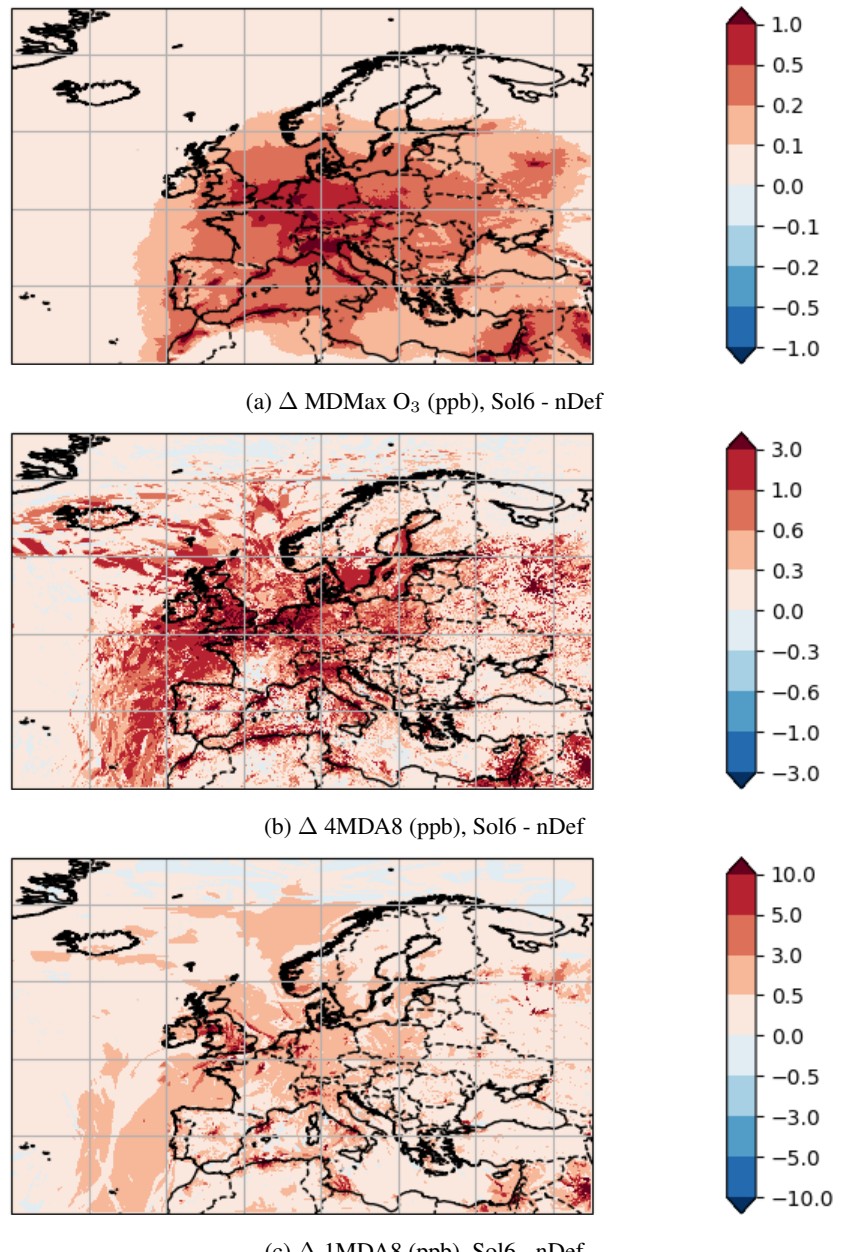

(a) Δ MDMax O$_3$ (ppb), Sol6 - nDef

(b) Δ 4MDA8 (ppb), Sol6 - nDef

(c) Δ 1MDA8 (ppb), Sol6 - nDef

**Figure G1.** Results of sensitivity tests with solvent speciation replaced by that of gasoline exhaust. Plots show differences (Sol6 - nDef, see text) for: (a) Mean of daily maximum O$_3$, (b) 4$^{th}$ highest MDA8 (c) highest daily MDA8.





(a) MDM O$_3$ (ppb)

(b) $\Delta$ MDMax O$_3$ (ppb)

(c) $\Delta$ 1MDA (ppb)

(d) $\Delta$ 4MDA (ppb)

(e) Emissions, NOx (mg/m$^2$)

(f) Emissions, NMVOC (mg/m$^2$)

**Figure G2.** Results of sensitivity tests for an area of Spain surrounding Madrid (marked with triangle): (a) base-case calculations of mean of daily maximum ceO3 (MDM O$_3$), (b) change in MDM O$_3$ when solvent speciation changed, (c) change in $4^{th}$ highest MDA8 when solvent speciation changed, (d) change in highest MDA8 when solvent speciation changed. Figs. (e) and (f) illustrate the emissions for NOx and NMVOC. Calculations with EmChem19 chemistry, CAMS-REG emissions, 0.1×0.1$^\circ$ resolution.