# Peer review of "Evaluation of modelled versus observed NMVOC compounds at EMEP sites in Europe"

_EGUsphere, 2023_

## Author Comment (AC1)

**egusphere-2023-3102: Evaluation of modelled versus observed NMVOC compounds at EMEP sites in Europe**

**Ge et al.**

**Response to Referee #1**

We thank the reviewer for their time spent reading our manuscript and for their recommendation for publication upon addressing their comments. Below we include all the reviewer's comments and provide in blue text our point-by-point responses. Please note that all section numbers and line numbers mentioned in our responses refer to the clean revised manuscript (not the track-changed version).

**General comments 1:**

Clarify the rationale for evaluating both 2018 and 2019, as the focus seems primarily on 2018 without clearly presenting key conclusions on the differences/similarities between years. Also, it should be important to clarify the selection of these years.

Response: One of the objectives of this study is to utilise the model in evaluating the 'goodness' of speciated emissions. To achieve this, we employed emission profiles from the UK National Atmospheric Emissions Inventory (NAEI). Our ideal scenario would involve conducting model simulations over several years to provide a better statistical basis for evaluating the robustness of our modelling results. However, this ideal is constrained in practice by the time available for this study, and the availability of speciated emission profiles, emission inventories, and ambient measurements. Firstly, hourly measurements prior to 2018 are scarce. Secondly, the latest speciated VOC emission profiles accessible from the NAEI is 2019. Thirdly, the latest sector-specific inventory provided by the Copernicus Atmosphere Monitoring Service (CAMS) is for 2018. Consequently, our model evaluation commenced in 2018, a year for which we had access to the CAMS inventory, the NAEI emission profile, and an adequate number of measurements. Regarding 2019, the modelling exhibited a performance comparable to that of 2018; therefore, to prevent unnecessary repetition, we only presented the results for 2018 in most cases.

To elucidate the differences/similarities across different years, the revised manuscript provides quantitative outcomes through Fig. 3 and 4, along with Tables 6 to 9. Additionally, the sentences in Sect. 3 (lines 315-320), Sect. 3.5 (lines 704-709), and Table 12 encapsulate a summary of these differences. Recognising that such information may have become obscured within the paper, we have taken steps to enhance clarity for our readers by adding the following lines.

**In Sect.2.5 lines 304-309 we added:**

"Owing to constraints in the availability of both emissions data and measurements, the analysis delineated herein mainly focuses on 2018, a year for which we had comprehensive access to the sector-specific CAMS inventory, the NAEI emission profile, and an adequate number of high-temporal-resolution (e.g., hourly) measurements. Model evaluations for 2019 and 2022 were carried out as supplementary activities, which were designed to make efficient use of both available regular monitoring and short-duration campaign measurements, providing additional evidence for robustness of our modelling results."

**In Sect.4 lines 764-765 we added:**

"For most species, the model's performance across 2018 and 2019 exhibits considerable similarity when evaluated against regular, year-round monitoring measurements."

**General comments 2:**

(Fig. 2) What could be behind the big differences between both inventories of the total VOC emissions of "C-Other combustion" and "D-Fugitive"? Looking at the speciation used, it seems that these sectors are important for some of the species evaluated in this study.

Response: We apologise for not specifying the domain differences between the two inventories in the initial version of the manuscript. In Figure 2 of the original manuscript, the raw data from both inventories were presented according to their respective domain ranges. To clarify, Table R1 outlines the boundary coordinates for each of the two inventories, as well as for the actual model domain utilised in our study. The observed discrepancies in the sector totals for "C-Other combustion" and "D-Fugitive" between the CEIP and CAMS inventories are due to the larger domain encompassed by CEIP. This results in generally higher reported totals by CEIP for these specific sectors.

**Table R1.** Coordinates of domain boundaries

|  | **Lon-west** | **Lon-east** | **Lat-south** | **Lat-north** |
|---|---|---|---|---|
| **CEIP** | -29.95 | 89.95 | 30.05 | 81.95 |
| **CAMS** | -29.95 | 59.95 | 30.02 | 71.97 |
| **Model domain** | -29.95 | 44.95 | 30.05 | 75.95 |

In the revised manuscript, we recalculated the total emissions for each sector based on the precise boundaries of the actual model domain, specifically longitude (-29.95 to 44.95) and latitude (30.05 to 75.95). Following this revision, the previously noted significant differences in the "C-Other combustion" and "D-Fugitive" sectors between the CEIP and CAMS inventories have largely disappeared. The specific model domain information is added to the caption of Fig. 2 (shown below) in the revised manuscript.

As a result, we have revised the descriptions of differences between the two inventories based on the values now calculated according to the actual model domain.

**In Sect. 2.3.4 we deleted the following sentence from the paragraph in lines 217:**

"The CAMS inventory, with its smaller domain, generally reports lower sector totals than the CEIP inventory."

**We have revised the texts in Sect. 3.5 lines 695-699 to the following:**

"For instance, the CEIP inventory identifies sector-E Solvents (24% of its total, similarly hereinafter) and sector-F Road Transport (22%) as major emitters (Fig. 2). In contrast, the CAMS inventory highlights Solvents as the most dominant sector (31%), which significantly surpasses other sectors. The second largest sector, Road Transport, which is further broken down into four sub-sectors each with their own distinct emission profiles, accounts for 15% of CAMS's annual total."

**We have revised the texts in Sect. 3.5 lines 702-710 to the following:**

"In general, model simulations using the CAMS inventory show slightly better agreements with measurements than those using the CEIP inventory, which is likely attributable to the detailed segmentation of the Road Transport sector within the CAMS inventory. For example, using the

CRIv2R5Em mechanism in 2018, the mean correlation coefficient is 0.59 for CEIP and 0.64 for CAMS. Moreover, both inventories result in model overestimation of n-butane and n-pentane but underestimation of i-butane and i-pentane, which is linked to the notably low i-to-n ratios of these species emanating from the solvent sector. Such findings imply that the emission profiles have considerable influence on the agreement between modelled and measured VOC concentrations, particularly for sectors with substantial emissions. Therefore, future focus to improve model accuracy may need to shift towards a more detailed breakdown of dominant emission sectors (e.g., Solvents) and the refinement of their speciation profiles."

**As a result, we have revised the texts in the Abstract lines 26-31 to the following:**

"Finally, model simulations employing the CAMS inventory show slightly better agreements with measurements than those using the CEIP inventory. This enhancement is likely due to the CAMS inventory's detailed segmentation of the Road Transport sector, including its associated subsector-specific emission profiles. Given this improvement, alongside the previously mentioned concerns about the model's biased estimations of various VOC ratios, future efforts should focus on a more detailed breakdown of dominant emission sectors (e.g., Solvents) and the refinement of their speciation profiles to improve model accuracy."

**Similarly in the Conclusions, we have revised the texts in lines 794-796 to the following:**

"Given this better model performance offered by CAMS, alongside the previously mentioned concerns about the model's biased estimations of various VOC ratios, future efforts should focus on a more detailed breakdown of dominant emission sectors (e.g., Solvents) and the refinement of their speciation profiles to improve model accuracy."

[Figure]

**Figure 2.** Annual total emissions (upper panel) from CAMS and CEIP inventories based on the same model domain (i.e., Longitude -29.95 to 44.95 and Latitude 30.05 to 75.95), and VOC profiles (lower panel) of individual EMEP sectors in CRIv2R5Em mechanism in 2018. Among the last 6 subsectors, PP stands for Public Power, RT stands for Road Transport. The speciation of sector F is an overall reflection of F1--F4. Note that CEIP does not provide data for the last six sectors (A1, A2, F1-F4), so emissions are zero for these sectors.

**General comments 3:**

Table 4 and Fig. 3 present the same information. I would remove/move to appendix one of them.

Response: Requested change made. Figure 3 is now moved to Appendix C as Fig. C1.

**General comments 4:**

The information provided in Section 4 feels out of place and should be introduced before. Table 5 should be moved to section 3.5 when presenting the model experiments. From L.293 to L.304, the information presented is related to the measurements and therefore should be moved to section 3.4.

Response: Requested change made. Table 5 is now moved to the end of Sect. 2.5.

**The following text related to the measurements in Sect. 4 (in the old manuscript) has been moved to Sect. 2.4 (in the revised manuscript) and now appears at lines 276-287 in the revised manuscript:**

"It is worth noting that the model-measurement comparison is complicated by the variation in the number of monitoring sites per species and in the frequency and duration of sampling time across stations. For example, the sampling duration for benzene varies from 5 to 40 minutes from DE0002R to GB0048R sites, while the model only calculates standard hourly concentrations. For this work we have matched the hourly model outputs with valid measurements at their native temporal resolution wherever we can. For instance, when using online Gas Chromatography (GC) measurements with an hourly resolution, such as CH0053R, we utilise the standard hourly model outputs. In contrast, for VOC measurements collected using the steel canister method (for example, FR0013R), these are compared with four-hour model averages (spanning 12:00 to 16:00) on the sampling day. This time frame is commonly used for canister sampling analysis, and the precise timing and duration of sampling within this time window often vary from one station to another. Therefore, due to the challenge in ascertaining these operational specifics for each station and species, we employ a model average over this period for comparison with the measured concentrations. Moreover, the annual mean concentrations discussed in this section are derived from hours with valid measurements, and where the sites have at least 65% data capture in a year."

**The remaining text is revised to the following, lines 317-321, to provide a preamble to the whole of Sect. 3:**

"This section provides a comparative analysis between modelled and measured surface VOC concentrations for the full years 2018 and 2019, and July 2022, using measurements from the standard EMEP monitoring network (Solberg et al., 2020) and 2022 IMP. The analyses for the years 2018 and 2019 reveal similar characteristics, and the model simulations employing varied mechanisms exhibit similar results as well. To avoid repetition, figures in this section are derived from the 2018 model simulation utilising the CRIv2R5Em mechanism alongside the CAMS inventory, except where indicated otherwise."

**General comments 5:**

(L.293-304) It is relevant to mention the measurements' time resolution. However, I would expect that not all the stations use the same measurement techniques. This should also lead to a certain degree of uncertainty when comparing the different stations. So, it would be interesting to also mention it.

Response: We thank the reviewer for making this useful suggestion.

**In Sect. 2.4 lines 288-296 we added:**

"Another factor adding complexity to the comparisons between model predictions and actual measurements is the variation in measurement techniques and the inherent analytical uncertainties associated with each method. For example, of the nine valid sites providing observations of ethyne, three utilise online gas chromatography (GC) and six employ steel canisters for sample collection coupled with offline GC. The former method uses continuous online monitors which offer hourly data, while the latter uses manual grab samples in canisters which essentially provide a snapshot measurement at specific time points, typically collected two to three times per week. Moreover, even within the same measurement technique, there are discrepancies in detection limits. For instance, the online GC at site CH0053R has a reported ethyne detection limit of 4.0 pmol/mol, in contrast to the online GC at site FI0096G, which has a significantly higher detection limit of 39.0 pmol/mol. These differences further underscore the challenges in achieving good model-measurement alignment when comparing data across different stations."

**General comments 6:**

In section 4.2.2, when compared to other sections, I was expecting more discussion on the results. For example, some comments on the station CH0053R.

Response: We thank the reviewer for their suggestion. To provide more information on the station CH0053R, we made a new plot, Fig. 13 in the revised manuscript, that shows the time series of modelled and measured aromatic species at this site in 2018.

**In Sect. 3.2.2 lines 523-537 we added:**

"In addition, the site CH0053R might initially appear as an anomaly, particularly in the comparisons of toluene and o-xylene. However, a detailed examination of the time series for these compounds at this site does not reveal any anomalies (as illustrated in Fig. 13). For benzene, there is good agreement between model predictions and measurements, with both indicating higher concentrations during winter and lower concentrations in summer, reflecting the expected seasonal variation.

In the case of toluene, the seasonal pattern is less pronounced. The measurements indicate several spikes in concentrations reaching 2 ppb during August and November, while the model suggests multiple peaks in April. Despite these discrepancies, for most of the year, the model and measurement data align reasonably well, both showing toluene concentrations fluctuating between 0.1 and 0.5 ppb.

Regarding o-xylene, the measured concentrations are consistently low throughout most of the year, typically close to zero, leading to significant analytical uncertainties. In comparison, the model predicts generally higher concentrations of o-xylene, with numerous peaks exceeding 0.05 ppb throughout the year. This discrepancy may arise from inaccuracies in the model's input data, suggesting a potential overestimation of emission sources within the single model grid considered. Nonetheless, considering the limited data available for this compound, and its low ambient levels, both model predictions and observed values are subject to considerable uncertainties. Consequently,

there is a need for more measurements focusing on not only air concentrations but also on emissions to enhance the accuracy of these estimates."

[Figure]

**Figure 13.** Time series comparisons of benzene, toluene, and o-xylene concentrations at CH0053R site in 2018.

**General comments 7:**

(L.550) The authors mention that ethyne and benzene share a similar source, but when looking at the speciation profiles (Fig.2) used for the different sectors I fail to see it. For example, the sector "C-other combustion" shows a big share for benzene but none for ethyne.

Response: This is exactly one of the key messages our paper wants to convey. The emission profile utilised in this work primarily originates from the UK National Atmospheric Emissions Inventory (NAEI), representing the most up-to-date speciated VOC inventory available to us. With these explicit emission inputs, we anticipated comparable model performance for ethyne as observed for benzene. Contrary to expectations, however, the model accurately predicts benzene levels yet struggles with ethyne. Besides, measurements (especially in winter) strongly indicate that ethyne and benzene originate from the same sources, a correlation the model fails to predict. This discrepancy underscores

that there are potentially significant flaws in the current estimates of ethyne emissions within the inventory, particularly in sectors with substantial benzene emissions, such as Other Combustion.

**General comments 8:**

(L.554) "....but the emission inputs need to be scrutinized and potentially revised to better reflect real-world conditions." How so?

Response: For instance, initiating measurement campaigns to assess ethyne emission factors, specifically targeting periods of high emissions such as winter, and focusing on proximate emission sources, including gasoline vehicles, shipping, industrial and residential combustion of natural gas, waste incineration, domestic wood fireplaces, could all yield more accurate data. The new winter VOC campaign currently being planned by the EMEP Task Force on Measurement and Modelling (TFMM) presents an excellent opportunity. However, more measurements will be required for different emission activities. These targeted campaigns would likely provide valuable insights into seasonal variations and the impact of specific sources on overall emissions.

**In Sect. 3.2.4 lines 585-590 we added:**

"For instance, initiating measurement campaigns to assess ethyne emission factors, specifically targeting periods of high emissions such as winter, and focusing on proximate emission sources, including gasoline vehicles, shipping, industrial and residential combustion of natural gas, waste incineration, domestic wood fireplaces, and so on. More emission measurements will be required for different emission activities. These targeted campaigns would likely provide valuable insights into seasonal variations and the impact of specific sources on overall emissions."

**General comments 9:**

The authors mention that the default speciation in the EMEP model is EmChem19rc, so why are the results presented mainly using CRIv2R5Em (L.305)? For the sensibility analysis, could the two chemical mechanisms show a very different sensitivity on ozone to changes in VOCs?

Response: The CRIv2R5Em mechanism contains a wider array of VOC species and more detailed chemistry compared to the EmChem19rc, thus providing an illustrative example of applying CRI schemes within the EMEP MSC-W model. One of the objectives of this study is to evaluate the performance of these two schemes, and to assess how sensitive VOC comparisons are to the choice of scheme, with details presented in Sect. 2.2 lines 105-122. The findings of this research suggest that the default EmChem19rc mechanism performs comparably to, if not better than, CRIv2R5Em, for selected VOC at least.

The decision to predominantly utilise CRIv2R5Em in this study was motivated by our aim to showcase results using the most comprehensive scheme available, which, in theory, should enhance model performance. However, it's important to note that no significant difference was observed between the two schemes in terms of their agreement with the relatively sparse VOC measurement data that are currently available. It's also important to note that running simulations with CRIv2R5Em incurs substantially higher computational costs than with EmChem19rc. For example, for a given number of high-performance computing (HPC) nodes, a full year's model simulation with CRIv2R5Em requires 1.5 times the run-time needed for EmChem19rc. While this increased computational demand is manageable for research purposes, it becomes impractical for routine operational modelling, which typically involves hundreds of model runs.

In other words, this research illustrates that the default EmChem19rc scheme, despite having a smaller set of VOC species and simpler chemistry, offers the advantages of speed and satisfactory accuracy. Consequently, it emerges as a great option for most EMEP modelling applications.

Regarding the second question about ozone sensitivity, the answer is no. The VOC comparisons detailed in this study already reveal that the two mechanisms exhibit remarkably similar behaviour. Additionally, Fig. R1 illustrates that both mechanisms yield ozone concentrations that align reasonably well with the observed hourly values at two example monitoring stations, with their results nearly indistinguishable from each other. This consistency in modelled VOC and ozone concentrations suggests that the oxidation processes offered by both mechanisms are very similar. Consequently, it is anticipated that the ozone concentrations modelled by either mechanism would demonstrate similar sensitivities when subjected to variations in VOC inputs.

[Figure]

**Figure R1.** Time series comparisons of ozone in 2018 between measurements (blue) and model outputs using the CRIv2R5Em (red) and EmChem19rc (green) mechanisms.

**General comments 10:**

I understand that the sensibility analysis presented in section 4.6 and Appendix G is beyond the scope of the study. My main concern goes towards the selection of the speciation profile. Why did the authors use a gasoline vehicle speciation profile to represent solvent emissions?

Response: The use of the gasoline profile was pragmatic - it was technically easy to implement, and anyway provided a more reactive VOC mixture than that of solvents. In particular, the gasoline profile contains more alkene species, which could be expected to influence short-term ozone formation. It is also clear (e.g. from Oliveira et al., 2023) that solvent speciation is extremely uncertain, and it is not known which alternative speciation best fits the European situation. We have modified the text to include this extra information.

**Technical corrections 1:**

Could be just a question of the formatting but some figures seem to appear misplaced in other subsections, e.g. Fig. 3 should be in section 3.4, Fig. 16 in section 4.3.2.

Response: We appreciate the reviewer highlighting the formatting issue. In the revised version, we have double-checked all figure placements and ensured they are in the correct subsections. For instance, Fig. 3 is now moved to the appendix as Figure C1, and Fig. 16 (now Fig. 15 in the revised manuscript) is now moved to section 3.3.2. We can note, however, it will be ACP typesetting that will decide on final placements; we can only make suggestions.

**Technical corrections 2:**

(L.608) "ose"?

Response: We apologise for the typo here.

**This sentence is rephrased to the following (lines 643-644):**

"The evaluation of model performance is inherently constrained by uncertainties in emissions, meteorological conditions, model parameterisation, and measurements."

**Technical corrections 3:**

Standardise, e.g., i-butane/iso-butane, i-pentane/iso-pentane, throughout the text.

Response: We thank the reviewer for pointing out these inconsistencies. Our intention had been to use "i" throughout our manuscript, since this terminology simplifies and standardises the expression of i-to-n ratios. However, some instances of "iso" escaped notice in the final read-through before submission. In the revised manuscript, we have checked again and corrected all "iso" to "i".

**Technical corrections 4:**

I leave the decision to the authors, but consider moving time series plots (Fig. 13 and 15) to the appendix.

Response: We thank the reviewer for this suggestion. Upon further consideration, we have opted to relocate the original Figure 13 to the Appendix F, where it now appears as Figure F2 in the revised manuscript. However, we have decided to maintain the original Figure 15 in its original position, where it is now labelled as Figure 14 in the revised manuscript.

---

## Author Comment (AC2)

**egusphere-2023-3102: Evaluation of modelled versus observed NMVOC compounds at EMEP sites in Europe**

**Ge et al.**

**Response to Referee #2**

We thank the reviewer for their time spent reading our manuscript. Below we include all the reviewer's comments and provide in blue text our point-by-point responses. Please note that the line numbers mentioned in our responses refer to the clean revised manuscript (not the track-changed version).

**Detailed comments 1:**

Line 10-12: I feel like the detailed configuration is not appropriate in the abstract.

Response: We thank the reviewer for their suggestion. We have now removed the specified sentences from the abstract.

**Detailed comments 2:**

Line 15-16: boundary conditions have the same impact for all species, why do you emphasize the impact for the latter species? Doesn't make sense. Please elaborate more.

Response: The EMEP model specifies the boundary and initial conditions (BICs) for various compounds, including VOCs, by employing a cosine function that describes month-to-month variations and takes into account latitude effects (as detailed in Sect. 3.3.6). Consequently, the impact of BICs on the concentrations of different VOC species varies.

For example, the consistent underestimation of propane concentrations by the model at all monitoring sites in both 2018 and 2019 can be partly attributed to issues with boundary conditions. This might involve an underestimation of the reference annual mean near-surface concentration, as determined from measurements at Mace Head, Ireland, or an underestimation of the seasonal cycle's amplitude, or potentially a combination of both factors. Moreover, significant spatial variations in propane concentrations at European background sites have been documented in multiple studies, as elaborated in Sect. 3.1.1, lines 332-339. This suggests that regional differences in BICs of propane may well play a role in the observed model-measurement discrepancies.

**To make the description of BICs more clear to readers, in Sect. 2.3.6 lines 253-254 we added:**

"Since different species have different annual means and amplitudes, the impact of BICs on the concentrations of different VOC species varies."

**Detailed comments 3:**

Line 24-25: In winter, the lifetime is longer. How will the bias of meteorology and chemistry affect the model performance?

Response: In general, biases in meteorology and chemistry are likely to affect all species uniformly. During winter, the lifetimes of ethene, benzene, and ethyne should become longer to a similar extent, implying that when examining ratios such as ethene-to-ethyne and benzene-to-ethyne, changes in their lifetimes should not significantly impact the results.

As detailed in Sect. 3.2.3 and Sect. 3.2.4, the key point is the model's strong agreement with the spatial correlations and time series for ethene and benzene measurements, but not for ethyne. This discrepancy highlights potential inaccuracies in ethyne emissions, given that all three compounds are primarily emitted from combustion-related activities. Notably, while the modelled ethyne concentrations align closely with measurements during summer, they diverge significantly in winter. In contrast, modelled concentrations of ethene and benzene consistently match observations across all seasons. More importantly, measurement data reveal a strong linear correlation between benzene and ethyne during winter across all sites, suggesting they share common emission sources. However, the model fails to predict this correlation. Given that ethyne and benzene have similar atmospheric lifetimes of 6-10 days (Table D1) and originate from similar human activities, such as fuel consumption and combustion processes, one would expect them to exhibit comparable spatial and temporal variation patterns. Yet, the observed satisfactory model performance for benzene contrasts sharply with the poor performance for ethyne, particularly during winter months. This suggests that the core issue lies in the accuracy of ethyne emission estimates.

**Detailed comments 4:**

Line 26: It's complicated. OVOC can also be emitted from biofuel combustion, etc.

Response: We thank the reviewer for their suggestion.

**To reflect the reviewer's suggestion, we have added this info to Sect. 3.3.2 lines 633-635:**

"Atmospheric sources of methylglyoxal are multiple and include direct emissions from, for example, industrial emissions, vehicle exhausts, biomass burning and biofuel combustion, and secondary formation from the oxidation of biogenic and anthropogenic precursors (e.g., isoprene, aromatics)."

**We also revised this sentence in the Abstract lines 23-26 to the following:**

"For OVOCs, the modelled and measured concentrations of methanal and methylglyoxal show a good agreement, despite a moderate underestimation by the model in summer. This discrepancy could be attributed to an underestimation of contributions from biogenic sources, or possibly a model overestimation of their photolytic loss in summer."

**Detailed comments 5:**

Line 29-30: I don't get the logic of this sentence. Since the performances are similar, how can you draw a conclusion on the significant impact of emission profiles?

Response: This sentence is linked to a very similar model performance that contrasts noticeably with the discrepancies between the two inventories. Following a re-evaluation of the differences in sector totals between the two inventories, as suggested by Reviewer 1, we have updated the upper panel of Figure 2 to reflect the actual model domain. Further details are provided in our response to Reviewer 1's General Comment 2.

**As a result, we have revised the text in the Abstract (lines 26-31) in line with these changes:**

"Finally, model simulations employing the CAMS inventory show slightly better agreements with measurements than those using the CEIP inventory. This enhancement is likely due to the CAMS inventory's detailed segmentation of the Road Transport sector, including its associated subsector-specific emission profiles. Given this improvement, alongside the previously mentioned concerns about the model's biased estimations of various VOC ratios, future efforts should focus on a more detailed breakdown of dominant emission sectors (e.g., Solvents) and the refinement of their speciation profiles to improve model accuracy."

**Detailed comments 6:**

Line 104: what's the difference between these two chemical mechanisms? Why do you choose these two?

Response: The EmChem19rc is the default chemical mechanism of the EMEP MSC-W model. The CRIv2R5Em is an EMEP adaption of the Common Representative Intermediates (CRI) v2-R5 mechanism. This mechanism is the simplest variant of CRI v2, considered suitable as a reference mechanism in large-scale chemistry-transport models. The difference between the two mechanisms is that CRIv2R5Em contains a wider array of VOC species and more detailed chemistry compared to the EmChem19rc (details in Sect. 2.2 lines 104-121), thus providing an illustrative example of applying CRI schemes within the EMEP MSC-W model.

The rationale behind selecting these two mechanisms was to assess the difference in model performance when employing either scheme. The results of this study indicate that the default EmChem19rc mechanism is on a par with CRIv2R5Em. It may also be noted that we predominantly present results from CRIv2R5Em in this study. This decision is driven by our aim to highlight findings using the most elaborate scheme available, which, theoretically, should enhance model performance. Nevertheless, it is crucial to mention that no significant difference was observed between the two schemes in terms of their agreement with measurements (at least as regards the measurement data available at this time). However, running simulations with CRIv2R5Em incurs substantially higher computational costs than with EmChem19rc. In other words, this research illustrates that the default EmChem19rc scheme, despite having a smaller set of VOC species and simpler chemistry, offers the advantages of speed and reasonable accuracy. Consequently, it emerges as a great option for most EMEP modelling applications.

**Detailed comments 7:**

Table 1: Don't understand to choose o-xylene as a tracer. They're active in chemical reaction, and can be produced as a secondary product. Why don't use benzene? Also, the same question for choosing C2H5OH as a tracer.

Response: In this context, the 'tracer' for o-xylene is coloured in green, signifying its creation for existing lumped surrogates. Benzene is explicitly simulated within the model, indicating it is processed based on its pure emissions and follows precise chemical reactions, thus eliminating the need for a tracer. Conversely, o-xylene and C2H5OH are themselves lumped surrogates within the model, which rely on aggregated emissions data. As a result, a tracer is necessary to obtain 'pure' concentrations that can be directly compared to ambient measurements.

**Detailed comments 8:**

Figure 4: Please improve this figure and the following ones. The labels are overlapped and hard to tell.

Response: We acknowledge the issue of label overlap and have explored various methods to present these data sets more clearly, including utilising station codes instead of full names, maintaining equal lengths for the x and y axes, and adjusting the axis aspect ratios of the plots in Python. Unfortunately, none of these approaches have successfully clarified every label for each data point. As can be observed, certain pairs of data points naturally cluster closely together, such as in the scatter plot for propane in 2018, where the differences in their modelled concentrations are very little, causing the data points to cluster near each other. Our chosen method of plotting, with equal x and y axes, provides the important benefit of positioning the one-to-one line at 45 degrees, thereby facilitating an easier assessment of the general trend regarding model overestimation or underestimation.

---

## Author Response (AR2)

Dear Professor West,

Thank you for your decision to publish our paper subject to some further minor revisions. Please find below our responses and actions with respect to your requests.

**Editor comment:**
Your responses to Referee #1 seems adequate and appropriate. For Referee #2, I am not so sure that your responses are adequate, and Referee #2 says that they are not willing to review the paper again. Rather than asking a different reviewer, I'll ask that you follow up on your responses to Referee #2. In particular:
1) "There are lots of information but less organized. The figure quality also needs to be improved." Please clarify what you've done to address this comment.

**Response:**
Unfortunately, Referee #2 did not provide any specific examples of how they believe our paper could be better organised. In addition, we point out that Referee #1 was entirely happy with the organisational aspect since they graded our paper "excellent" for the review question "Are the scientific results and conclusions presented in a clear, concise, and well-structured way?". There is therefore the risk that in trying to second-guess the concern of Referee #2 we make modifications that would degrade the presentation experience for Ref #1, ourselves, and other readers of our paper. Having said that, during the revision process we did make a few presentational changes that we believe both slightly shorten and enhance the paper's organisation: specifically, we moved both the original Figures 3 and 13 to the Supplementary; we moved some text that related to measurements that was previously in a result section into the methods section; we moved the original Table 5 to the end of Section 2.5; and we provided a revised overview preamble to the whole results section.

Our action concerning the comment on clarity of some of the figures is given in the response that follows.

**Editor comment:**
2) Regarding several figures and the comment "the labels are overlapped and hard to tell" you state that you tried to create the figures differently but were not successful. I agree that these plots look unprofessional and are hard to interpret. There are not so many points that I would think you could change the location of some labels by hand using some figure editing software to make it more clear, even if you are not able to do so through your plotting software. Please consider again whether such changes might be able to improve the figure.

**Response:**
This comment relates to the attempt to include individual site labels next to each data point on scatter plots of observed vs modelled concentrations for a given atmospheric species. We firmly wish to retain figure panels that have equal x and y axis so that the line of observed and modelled equality is at an intuitive 45 degrees through the panel. However, for some species the concentrations vary very little from site to site causing many data points to overlap or be very close to each other. In most of these scatter plots there are too many closely spaced data points to fit in labels for all of them as well. Our solution is therefore to remove all labels except for those marking data points that are specifically discussed in the text. Namely, Figure 3, 4, 8, 11, 12, and 15 in the main paper are all updated to reflect this revision. All the site names and their respective data values for each figure panel are now included in tables in the Supplement (i.e., Tables E1, E2, E3, F1, F2, F3, G1) so that readers have access to all the quantitative information. The corresponding Supplementary table numbers have now been added to each figure caption.

To illustrate, Figure 11 in the revised manuscript Sect. 3.2.1 (also provided below for your reference) omits all data labels except for DE0007R for isoprene, an outlier that significantly weakens the linear correlation.

[Figure]

Figure 11. Scatter plots of annual mean modelled and measured ethene, ethyne, and isoprene concentrations in 2018. The term 'CRI' indicates that the model data is calculated using the CRIv2R5Em mechanism. In each plot, the grey line is the 1:1 line, and the other coloured line is the least-squares regression line. For isoprene, the outlier site is plotted in red; the red line is the regression line with the outlier; the green line is the regression line without the outlier. The site codes and their respective data values for each figure panel are provided in Supplement Table F1.

**Editor comment:**
3) For Referee #2, please consider whether you can make changes to the text of the paper to clarify the issues raised regarding comments #3, 6 and 7. Your written response to the comment makes sense, but no changes were made to the text to clarify these issues for other readers.

**Response:**
The referee's comment #3 concerns potential bias of meteorology and chemistry affecting the model performance; we provided a comprehensive response to this in our response document.

To further clarify this for other readers, we have added the following text to the main paper Sect. 3.2.4, **lines 581-584**:

"In general, biases in meteorology and chemistry are likely to affect all species uniformly. During winter, the lifetimes of ethene, benzene, and ethyne should become longer to a similar extent, implying that when examining ratios such as ethene-to-ethyne and benzene-to-ethyne, changes in their lifetimes should not significantly impact the results."

The remainder of our original response to the referee's comment #3 is essentially a shorter version of the paragraph also in Sect. 3.2.4 between lines 579-594. To further concentrate the message while avoiding repetition, we have added the following text to the end of this section as a summary, appearing in **lines 615-621**:

"In summary, the key difference is the model's strong agreement with the spatial correlations and time series for ethene and benzene measurements, but not for ethyne. While the modelled ethyne concentrations align closely with measurements during summer, they diverge significantly in winter. In contrast, modelled concentrations of ethene and benzene consistently match observations across all seasons. More importantly, measurement data reveal a strong linear correlation between ethene and ethyne, and between benzene and ethyne, during winter across all sites, suggesting they share common emission sources. However, the model fails to predict this correlation. This discrepancy highlights potential inaccuracies in ethyne emissions, given that all three compounds are commonly emitted from combustion-related activities."

The referee's comment #6 queries the rationale for the choice of the gas-phase chemical mechanisms investigated in our model simulations. Our original response to this comment consists of 2 paragraphs. The first paragraph is essentially a copy-and-paste of sentences in Sect. 2.2 lines 104-121. The second paragraph explains the rationale behind selecting these two mechanisms. To make this message clearer to other readers while avoiding repetition, we have selectively added the following sentences to the revised manuscript Sect. 2.2 **lines 141-151**:

"In summary, the difference between the two mechanisms is that CRIv2R5Em contains a wider array of VOC species and more detailed chemistry compared to the EmChem19rc, thus providing an illustrative example of applying CRI schemes within the EMEP MSC-W model. The rationale behind selecting these two mechanisms was to assess the difference in model performance when employing either scheme. The results of this study (Sect. 3) indicate that the default EmChem19rc mechanism is on a par with CRIv2R5Em. We mainly present results from CRIv2R5Em in this study because we aim to highlight findings using the most elaborate scheme available, which, theoretically, should enhance model performance. Nevertheless, it is crucial to mention that no significant difference was observed between the two schemes in terms of their agreement with measurements at least as regards the measurement data available at this time. However, running simulations with CRIv2R5Em incurs substantially higher computational costs than with EmChem19rc. In other words, this research illustrates that the default EmChem19rc scheme, despite having a smaller set of VOC species and simpler chemistry, offers the advantages of speed and reasonable accuracy. "

The referee's comment #7 suggests some misunderstandings concerning the use of o-xylene as a tracer. To also clarify this for other readers, we have added the following sentences in Sect. 2.2 **lines 136-140** to explain the tracer system using o-xylene as an example:

"For example, benzene is explicitly simulated within the model, meaning that it is processed based on its own individual emissions, thus eliminating the need for a tracer. Conversely, o-xylene is itself a lumped surrogate within the model, which relies on aggregated emissions data. As a result, a tracer OXYL_T is necessary to obtain 'pure' concentrations that can be directly compared to ambient measurements."